# Preconditioned Riemannian Gradient Descent Algorithm for Low-Multilinear-Rank Tensor Completion

**Yuanwei Zhang** [1]  **Fengmiao Bian** [2]  **Xiaoqun Zhang** [1 3]  **Jian-Feng Cai** [2]

## Abstract

Tensors play a crucial role in numerous scientific and engineering fields. This paper addresses the low-multilinear-rank tensor completion problem, a fundamental task in tensor-related applications. By exploiting the manifold structure inherent to fixed-multilinear-rank tensor set, we introduce a simple yet highly effective preconditioned Riemannian metric and propose the Preconditioned Riemannian Gradient Descent (PRGD) algorithm. Compared to the standard Riemannian Gradient Descent (RGD), PRGD achieves faster convergence while maintaining the same order of per-iteration computational complexity. Theoretically, we provide the recovery guarantee for PRGD under near-optimal sampling complexity. Numerical results highlight the efficiency of PRGD, outperforming state-of-the-art methods on both synthetic data and real-world video inpainting tasks. Code is available at https://github.com/Jiushanqing-0418/PRGD-Tucker.

## 1. Introduction

Tensors are multidimensional arrays, which are routinely arising in different fields, such as community detection (Jing et al., 2021; Zhen & Wang, 2023), multi-dimensional recommendation system (Zhou et al., 2024), video and hyperspectral image processing (Zhang et al., 2021; Yang et al., 2023), signal processing (Chang & Wu, 2022) and quantum state tomography (Qin et al., 2024). In this paper, we consider the tensor completion problem, where the goal is to reconstruct the underlying three-order tensor $\mathcal{X}_* \in \mathbb{R}^{n_1 \times n_2 \times n_3}$ from its partially observed entries:

$$\mathcal{X}_*(i,j,k), \quad (i,j,k) \in \Omega,$$

where $\Omega$ is the index set of observed entries. However, without any assumptions, the problem is ill-posed since the number of observed entries $|\Omega|$ is usually much smaller than the ambient dimension $n_1 n_2 n_3$. One commonly adopted assumption is the low-rankness of $\mathcal{X}_*$. Compared to matrices, tensors possess a richer variety of rank notions, each associated with different tensor decomposition structures. Examples include the CP decomposition (Hitchcock, 1927), Tucker decomposition (Tucker, 1966), Tensor-Train decomposition (Oseledets, 2011) and several others (Kolda & Bader, 2009). In this work, we focus on the Tucker decomposition and its associated multilinear rank. The low-multilinear-rank tensor completion problem is formulated as follows:

$$\min_{\mathcal{X} \in \mathbb{R}^{n_1 \times n_2 \times n_3}} f(\mathcal{X}) := \frac{1}{2} \|\mathscr{P}_\Omega(\mathcal{X}) - \mathscr{P}_\Omega(\mathcal{X}_*)\|_F^2 \tag{1}$$
$$\text{s.t. } \operatorname{rank}(\mathcal{X}) = \boldsymbol{r},$$

where $\boldsymbol{r} = (r_1, r_2, r_3)$ is the multilinear rank satisfies $r_i \le n_i$ for all $i = 1, 2, 3$, $\mathscr{P}_\Omega$ is a projection operator for element-wisely sampling and $\|\cdot\|_F$ denotes the Frobenius norm.

### 1.1. Related Works

Compared with low-rank matrix completion, low-rank tensor completion presents unique challenges due to its more sophisticated algebraic structure. For instance, while nuclear norm minimization achieves near-optimal sampling complexity for low-rank matrix completion (Candès & Tao, 2010), computing the tensor nuclear norm is NP-hard (Friedland & Lim, 2018) and applying nuclear norm minimization on tensor's unfolding matrices results in suboptimal sampling complexity $O(n^2)$ (where $n = \max_{i=1,2,3} n_i$ and assume $r_i = O(1), i = 1, 2, 3$) (Mu et al., 2014; Huang et al., 2015). The state-of-the-art sampling complexity of polynomial-time algorithms is $O(n^{3/2})$ (Cai et al., 2022b; Tong et al., 2022; Wang et al., 2023), which is considered near-optimal given the conjecture that no polynomial-time

[1]School of Mathematical Sciences and Institute of Natural Sciences, Shanghai Jiao Tong University, Shanghai, China. [2]Department of Mathematics, The Hong Kong University of Science and Technology, Hong Kong, China. [3]Shanghai Artificial Intelligence Laboratory, Shanghai, China. Correspondence to: Jian-Feng Cai <jfcai@ust.hk>, Xiaoqun Zhang <xqzhang@sjtu.edu.cn>.

*Proceedings of the 42ⁿᵈ International Conference on Machine Learning*, Vancouver, Canada. PMLR 267, 2025. Copyright 2025 by the author(s).

algorithm can achieve a sampling complexity lower than this threshold (Barak & Moitra, 2016). These algorithms can be broadly categorized into two main types.

The first category consists of algorithms based on the factorization form of the Tucker format, where the optimization variables are factor matrices and the core tensor. Among them, (Xia & Yuan, 2019) assumed orthogonality of the factor matrices and proposed a Grassmannian gradient descent algorithm. (Han et al., 2022) introduced regularization to enforce orthogonal constraints on factor matrices, resulting in a regularized gradient descent algorithm. (Tong et al., 2022) developed a scaled gradient descent algorithm and derived an iteration complexity that is independent of $\mathcal{X}_*$'s condition number.

Another category involves algorithms that optimize directly on the tensor variable and employ projection in each iteration to exploit the low-rank structure. (Rauhut et al., 2017) developed an iterative hard thresholding algorithm and (Chen et al., 2019) proposed a projected gradient descent algorithm. Among those approaches, Riemannian optimization methods are particularly noteworthy due to their theoretical advantages and computational efficiency. For instance, (Wang et al., 2023) developed the Riemannian gradient descent algorithm and established the entrywise linear convergence guarantee. (Luo & Zhang, 2023) proposed a Riemannian Gaussian-Newton algorithm that achieves a quadratic convergence rate. Notably, Riemannian optimization methods also exhibit an iteration complexity that is free of the condition number (Cai et al., 2022b; 2023).

To the best of our knowledge, existing Riemannian optimization methods developed on the fixed-multilinear-rank manifold for (1) exclusively utilize the canonical metric induced by the ambient space $\mathbb{R}^{n_1 \times n_2 \times n_3}$. However, the Riemannian metric can vary across points on the manifold, providing the potential for more tailored designs. Thus, a natural question arises:

*Can we construct a more suitable Riemannian metric that enables the resulting Riemannian gradient descent algorithm for (1) to achieve faster convergence while maintaining near-optimal sampling complexity?*

Preconditioning techniques have long been recognized as powerful tools to accelerate convergence. Our work contributes to the recent strand of works on preconditioned methods for tensor/matrix variable optimization problems (Kasai & Mishra, 2016; Gupta et al., 2018; Tong et al., 2022; Cai et al., 2022a; Gao et al., 2024; Bian et al., 2024; 2025). Notably, (Gupta et al., 2018) proposed a preconditioned method, called Shampoo, for unconstrained tensor optimization problems. This method constructs preconditioners by accumulating outer products of historical gradients. However, Shampoo's preconditioners are fully dense, rendering them computationally expensive for large-scale problems. In this work, we develop a simple yet efficient preconditioner along with a data-driven Riemannian metric on the tensor manifold, resulting in the PRGD algorithm for (1). Numerical experiments on synthetic data show that the proposed PRGD algorithm can be several times faster than state-of-the-art algorithms. Additionally, we provide a theoretical recovery guarantee, demonstrating that PRGD achieves near-optimal sampling complexity.

## 1.2. Our Contributions

The main contributions of this work are summarized as follows:

- *Data-driven Riemannian metric and tangent space parameterization.* We design a computationally efficient, data-driven metric from the gradient and equip the tangent space of the iterate on the manifold with this data-driven metric. To compute the induced Riemannian gradient, we introduce a novel tangent space parameterization along with a detailed analytical framework, providing deeper insights into this method.

- *PRGD algorithm and empirical studies.* Building on the data-driven metric and induced Riemannian gradient, we proposed the PRGD algorithm and empirical studies on both synthetic and real datasets. The computational complexity of PRGD maintains the same order as RGD's per iteration. The numerical results demonstrate that PRGD can be several times faster than state-of-the-art algorithms.

- *Recovery guarantee.* We analyze the theoretical properties of PRGD and establish that, with proper initialization, such as using spectral methods, PRGD converges linearly to the global minimizer with a contracting factor that is independent of the condition number. Furthermore, PRGD achieves a near-optimal sampling complexity of $O(n^{3/2})$.

**Notations.** In this paper, we use the calligraphic letters $(\mathcal{X}, \widehat{\mathcal{X}})$ to denote tensors, bold-face capital letters $(\boldsymbol{U}, \widehat{\boldsymbol{U}})$ to denote matrices, lower case bold-face letters $(\boldsymbol{x}, \boldsymbol{y})$ to denote vectors, mathematical scripts $(\mathscr{P}, \mathscr{I})$ to denote operators, and blackboard bold-face letters $(\mathbb{R}, \mathbb{M})$ to denote sets. Additionally, for matrix $\boldsymbol{U} \in \mathbb{R}^{m_1 \times m_2}$, we define $\|\boldsymbol{U}\|_{2,\infty} := \max_k \|\boldsymbol{U}(k,:)\|_2$ is the $\ell_{2,\infty}$ norm of $\boldsymbol{U}$. We define $\bar{n} = (n_1 n_2 n_3)^{1/3}$ as the geometric mean, $n = \max_{i=1,2,3} n_i$ and $r = \max_{i=1,2,3} r_i$. For constant $a$ and $b$, $a \vee b = \max\{a, b\}$ and $a \wedge b = \min\{a, b\}$.

## 2. Preliminaries

### 2.1. Tensor Operations and Tucker Format

We give a brief introduction of tensor notations here for the paper to be self-contained, interested readers can refer to (Kolda & Bader, 2009) for more details.

*Tensor matricization and multiplication.* For tensor $\mathcal{X} \in \mathbb{R}^{n_1 \times n_2 \times n_3}$, the mode-1 matricization $\mathscr{M}_1(\mathcal{X}) \in \mathbb{R}^{n_1 \times n_2 n_3}$ is given by $[\mathscr{M}_1(\mathcal{X})]_{i_1,i_2,i_3} = \mathcal{X}_{i_1,(i_2-1)n_2+i_3}$ for $i_k \in [n_k]$. The $\mathscr{M}_2(\mathcal{X})$ and $\mathscr{M}_3(\mathcal{X})$ are defined similarly. We denote $\times_i$ as the the mode-$i$ tensor multiplication, for matrix $\boldsymbol{U} \in \mathbb{R}^{m \times n_1}$, the mode-1 multiplication $\mathcal{X} \times_1 \boldsymbol{U} \in \mathbb{R}^{m \times n_2 \times n_3}$ satisfies

$$\mathscr{M}_1(\mathcal{X} \times_1 \boldsymbol{U}) = \boldsymbol{U}\mathscr{M}_1(\mathcal{X}).$$

*Multilinear rank and Tucker format.* The multilinear rank of $\mathcal{X}$ is the collection $\text{rank}(\mathcal{X}) := (\text{rank}(\mathscr{M}_1(\mathcal{X})), \text{rank}(\mathscr{M}_2(\mathcal{X})), \text{rank}(\mathscr{M}_3(\mathcal{X})))$. If $\mathcal{X}$ has multilinear rank $\boldsymbol{r} = (r_1, r_2, r_3)$, then there exist $\mathcal{D} \in \mathbb{R}^{r_1 \times r_2 \times r_3}$ and $\boldsymbol{U}_i \in \mathbb{R}^{n_i \times r_i}$ satisfying $\boldsymbol{U}_i^\top \boldsymbol{U}_i = \boldsymbol{I}_{r_i}$ for $i = 1, 2, 3$ such that

$$\mathcal{X} = \mathcal{D} \times_{i=1}^3 \boldsymbol{U}_i.$$

This is called the Tucker format of $\mathcal{X}$.

*Tensor condition number.* For $\mathcal{X}$ with multilinear rank $\boldsymbol{r} = (r_1, r_2, r_3)$. The condition number of $\mathcal{X}$ is defined as

$$\kappa := \frac{\sigma_{\max}(\mathcal{X})}{\sigma_{\min}(\mathcal{X})} := \frac{\max_{i=1,2,3} \sigma_1(\mathscr{M}_i(\mathcal{X}))}{\min_{i=1,2,3} \sigma_{r_i}(\mathscr{M}_i(\mathcal{X}))},$$

where $\sigma_r(\cdot)$ is the $r$-th largest singular value of input matrix.

*Tensor norm.* For tensor $\mathcal{X}, \mathcal{Y} \in \mathbb{R}^{n_1 \times n_2 \times n_3}$, the inner product between two tensors is defined as

$$\langle \mathcal{X}, \mathcal{Y} \rangle = \sum_{i,j,k} \mathcal{X}(i,j,k) \cdot \mathcal{Y}(i,j,k).$$

The Frobenius norm is defined as $\|\mathcal{X}\|_F = \sqrt{\langle \mathcal{X}, \mathcal{X} \rangle}$.

### 2.2. Incoherence Condition

For the completion problem, it is crucial to assume the information of $\mathcal{X}^*$ carries fairly among all its entries to make the problem well-posed. The following two concepts are widely studied to characterise such property (Yuan & Zhang, 2016; Cai et al., 2023).

**Definition 2.1** (Incoherence and Spikiness)**.** For $\mathcal{X}_*$ with multilinear rank $\boldsymbol{r} = (r_1, r_2, r_3)$ and $\mathcal{X}_* = \mathcal{D}_* \times_{i=1}^3 \boldsymbol{U}_i^*$. The the incoherence and spikiness parameter of $\mathcal{X}_*$ are defined as:

$$Incoh(\mathcal{X}_*) := \max_{i=1,2,3} \sqrt{\frac{n_i}{r_i}} \|\boldsymbol{U}_i^*\|_{2,\infty},$$

and

$$Spiki(\mathcal{X}_*) := \frac{\|\mathcal{X}_*\|_{\ell_\infty}}{\|\mathcal{X}_*\|_F} \sqrt{n_1 n_2 n_3}.$$

A low incoherence or spikiness parameter indicates that the entries of the tensor are evenly distributed, enabling a small subset of entries to effectively capture and reveal the structure of the entire tensor. The relationship between these two parameters is as follows.

**Lemma 2.2** (Lemma 13.5. (Cai et al., 2023))**.** *Let* $\mathcal{X}_*$ *with multilinear rank* $\boldsymbol{r} = (r_1, r_2, r_3)$*,* $Incoh(\mathcal{X}_*) = \mu$ *and* $Spiki(\mathcal{X}_*) = \nu$*, then*

$$\mu \leq \nu \kappa_0,$$

*where* $\kappa_0$ *is the condition number of* $\mathcal{X}_*$*.*

### 2.3. Fixed-Multilinear-Rank Manifold and RGD Algorithm

For tensors in $\mathbb{R}^{n_1 \times n_2 \times n_3}$, the set of tensors with multilinear rank $\boldsymbol{r} = (r_1, r_2, r_3)$ forms a smooth embedded submanifold, denoted as $\mathbb{M}_{\boldsymbol{r}}$, i.e.

$$\mathbb{M}_{\boldsymbol{r}} = \{\mathcal{X} \in \mathbb{R}^{n_1 \times n_2 \times n_3} | \text{rank}(\mathcal{X}) = \boldsymbol{r}\}.$$

Given $\mathcal{X} \in \mathbb{M}_{\boldsymbol{r}}$, $\mathcal{X} = \mathcal{D} \times_{i=1}^3 \boldsymbol{U}_i$ with $\boldsymbol{U}_i^\top \boldsymbol{U}_i = \boldsymbol{I}_{r_i}$. Denote $\mathbb{T}_{\mathcal{X}}$ the tangent space of $\mathcal{X}$, given tensor $\mathcal{Z}$, the projection has the form

$$\mathscr{P}_{\mathbb{T}_{\mathcal{X}}}(\mathcal{Z}) = \mathcal{C} \times_{i=1}^3 \boldsymbol{U}_i + \sum_{i=1}^3 \mathcal{D} \times_i \boldsymbol{W}_i \times_{j \neq i} \boldsymbol{U}_j,$$

where $\mathcal{C}$ and $\boldsymbol{W}_i$ are parameters computed via explicit formulas by $\mathcal{D}, \boldsymbol{U}_i$, and $\mathcal{Z}$. By equipping $\mathbb{M}_{\boldsymbol{r}}$ with the canonical metric, it becomes a Riemannian manifold. Riemannian optimization method, recognized for their computational efficiency and sampling optimality, have been extensively studied for solving (1) (Kressner et al., 2014; Wang et al., 2023; Luo & Zhang, 2023; Zhang et al., 2024). The Riemannian gradient descent (RGD) algorithm iteratively generates a sequence as follows:

$$\mathcal{X}^{t+1} = \mathscr{R}_t \left( \mathcal{X}^t - \alpha_t \cdot \text{grad} f(\mathcal{X}^t) \right), \qquad (2)$$

where $\text{grad} f(\mathcal{X}^t)$ is the Riemannian gradient at $\mathcal{X}^t$, $\alpha_t$ is the stepsize, and $\mathscr{R}_t$ is the retraction operator. The iteration (2) contains the following two key components:

- *Riemannian gradient and tangent space projection.* The Riemannian gradient depends on the metric used in tangent space. Under the canonical metric, $\text{grad} f(\mathcal{X}^t) = \mathscr{P}_{\mathbb{T}_{\mathcal{X}^t}} \nabla f(\mathcal{X}^t)$, where $\nabla f$ is the Euclidean gradient of $f$ in the space $\mathbb{R}^{n_1 \times n_2 \times n_3}$. By projecting the Euclidean gradient onto $\mathbb{T}_{\mathcal{X}^t}$, the Riemannian gradient is at most rank $2\boldsymbol{r}$, rendering considerable computational efficiency in subsequent retraction step (Kressner et al., 2014; Cai et al., 2020).

- *Retraction by HOSVD.* The retraction operator $\mathscr{R}_t$ is a map from tangent space $\mathbb{T}_{\mathcal{X}^t}$ to manifold $\mathbb{M}_r$. A widely used choice for retraction is the high-order singular value decomposition (HOSVD) operator $\mathscr{H}_r$ (De Lathauwer et al., 2000). For $\mathcal{Z} \in \mathbb{R}^{n_1 \times n_2 \times n_3}$,

$$\mathscr{H}_r(\mathcal{Z}) := \mathcal{Z} \times_{i=1}^3 \boldsymbol{Z}_i \boldsymbol{Z}_i^\top,$$

where $\boldsymbol{Z}_i \in \mathbb{R}^{n_i \times r_i}$ is the $r_i$-principle left singular vector matrix of $\mathscr{M}_i(\mathcal{Z})$.

The efficiency of RGD is significantly influenced by the choice of the Riemannian metric. This motivates the central aim of this work: designing a more suitable Riemannian metric on $\mathbb{M}_r$ to achieve faster convergence.

## 3. Preconditioned RGD

### 3.1. Data-Driven Metric and Riemannian Gradient

Our preconditioners are derived from the gradient at each iteration, yielding a data-driven metric tailored to the given observations and algorithmic data. Recognizing the importance of incoherence and spikiness in tensor completion, our preconditioner rescales the gradient entry-wise to enhance these properties.

Let $\mathcal{G}^t := \mathscr{P}_\Omega(\mathcal{X}^t) - \mathscr{P}_\Omega(\mathcal{X}_*) \in \mathbb{R}^{n_1 \times n_2 \times n_3}$ be the Euclidean gradient of the objective function $f$ at iterate $\mathcal{X}^t$. We define the following diagonal matrices for $i = 1, 2, 3$:

$$\boldsymbol{G}_{t,i} = \left(\epsilon_t \cdot \boldsymbol{I}_{n_i} + \mathscr{P}_{\mathrm{diag}}(\mathscr{M}_i(\mathcal{G}^t)\mathscr{M}_i(\mathcal{G}^t)^\top)\right)^{\frac{1}{6}}, \quad (3)$$

where $\epsilon_t > 0$ is a constant and $\mathscr{P}_{\mathrm{diag}}(\cdot)$ sets the off-diagonal elements of input matrix to zeros, making $\boldsymbol{G}_{t,i}$ positive definite diagonal matrices. The exponent $\frac{1}{6}$ is applied entrywise to the diagonal vectors and the value 6 is set twice the tensor order for normalization. We then define a new inner product in $\mathbb{R}^{n_1 \times n_2 \times n_3}$ as follows:

$$\langle \mathcal{Y}, \mathcal{Z} \rangle_{\mathscr{W}_t} := \langle \mathscr{W}_t(\mathcal{Y}), \mathcal{Z} \rangle = \langle \mathcal{Y} \times_{i=1}^3 \boldsymbol{G}_{t,i}, \mathcal{Z} \rangle. \quad (4)$$

where $\mathcal{Y}, \mathcal{Z} \in \mathbb{R}^{n_1 \times n_2 \times n_3}$ are two arbitrary tensors. It is straightforward to verify that $\langle \cdot, \cdot \rangle_{\mathscr{W}_t}$ is an inner product. This weighted inner product is then endowed in the tangent space $\mathbb{T}_{\mathcal{X}^t}$, and our preconditioned Riemannian gradient descent is derived under this metric.

Notice that $\boldsymbol{G}_{t,i}$ is a diagonal matrix whose diagonals are approximately the norms of mode-$i$ slices of $\mathcal{G}^t$. Therefore, in the preconditioned Riemannian gradient descent algorithm, each element of $\mathcal{G}^t$ is scaled by the inverse of the product of its slice norms, which unifies its entries and enhances the incoherence condition. We note that our preconditioner is also closely related to Shampoo's preconditioners (Gupta et al., 2018), but we discard historical gradients and retain only the diagonals of the preconditioner matrices, thereby

enhancing computational and memorial efficiency. Similar preconditioner has also been developed in (Bian et al., 2024; 2025).

**Proposition 3.1.** *For $\mathcal{X} \in \mathbb{M}_r$, denote $\widetilde{\mathscr{P}}_{\mathbb{T}_\mathcal{X}}$ as the projection operator onto $\mathbb{T}_\mathcal{X}$ under $\langle \cdot, \cdot \rangle_{\mathscr{W}_t}$ and $\mathcal{G}$ is the Euclidean gradient at $\mathcal{X}$. Then the Riemannian gradient $\widetilde{\mathrm{grad}} f(\mathcal{X})$ under our data-driven metric is:*

$$\widetilde{\mathrm{grad}} f(\mathcal{X}) = \widetilde{\mathscr{P}}_{\mathbb{T}_\mathcal{X}} \mathscr{W}_t^{-1} \mathcal{G}.$$

However, it is difficult to directly obtain the explicit form of $\widetilde{\mathscr{P}}_{\mathbb{T}_\mathcal{X}}$ since the components in standard tangent space parameterization are not orthogonal under our data-driven metric $\langle \cdot, \cdot \rangle_{\mathscr{W}_t}$. We need to reparameterize the tangent space $\mathbb{T}_\mathcal{X}$ according to our data-driven metric.

### 3.2. Tangent Space Parameterization

We first give the following lemma and the proof can be found in the Appendix A.

**Lemma 3.2.** *Given a gauge sequence $\boldsymbol{G}_i \in \mathbb{R}^{n_i \times n_i}, i = 1, 2, 3$ (e.g. each $\boldsymbol{G}_i$ is a symmetric positive definite matrix). For $\mathcal{X} \in \mathbb{M}_r$ with $\mathcal{X} = \mathcal{D} \times_{i=1}^3 \widetilde{\boldsymbol{U}}_i$ and $\widetilde{\boldsymbol{U}}_i^\top \boldsymbol{G}_i \widetilde{\boldsymbol{U}}_i = \boldsymbol{I}_{r_i}, i = 1, 2, 3$. Then each element in the tangent space at $\mathcal{X}$ can be represented as:*

$$\mathcal{C} \times_{i=1}^3 \widetilde{\boldsymbol{U}}_i + \sum_{i=1}^3 \mathcal{D} \times_i \widetilde{\boldsymbol{W}}_i \times_{j \neq i} \widetilde{\boldsymbol{U}}_j \quad (5)$$

*with $\widetilde{\boldsymbol{W}}_i \in \mathbb{R}^{n_i \times r_i}$, $\widetilde{\boldsymbol{U}}_i^\top \boldsymbol{G}_i \widetilde{\boldsymbol{W}}_i = \boldsymbol{0}_{r_i}, i = 1, 2, 3$ and $\mathcal{C} \in \mathbb{R}^{r_1 \times r_2 \times r_3}$.*

Note that each $\boldsymbol{G}_{t,i} \in \mathbb{R}^{n_i \times n_i}$ is positive definite. So we can use $\boldsymbol{G}_{t,i}$'s as a gauge sequence in Lemma 3.2 and obtain the following proposition.

**Proposition 3.3.** *For $\mathcal{X} \in \mathbb{M}_r$, use $\boldsymbol{G}_{t,i}$'s as a gauge sequence in Lemma 3.2, then the four components in (5) are orthogonal to each other under $\langle \cdot, \cdot \rangle_{\mathscr{W}_t}$:*

$$\langle \mathcal{C} \times_{i=1}^3 \widetilde{\boldsymbol{U}}_i, \mathcal{D} \times_i \widetilde{\boldsymbol{W}}_i \times_{j \neq i} \widetilde{\boldsymbol{U}}_j \rangle_{\mathscr{W}_t} = 0, i = 1, 2, 3,$$
$$\langle \mathcal{D} \times_k \widetilde{\boldsymbol{W}}_k \times_{j \neq k} \widetilde{\boldsymbol{U}}_j, \mathcal{D} \times_l \widetilde{\boldsymbol{W}}_l \times_{j \neq l} \widetilde{\boldsymbol{U}}_j \rangle_{\mathscr{W}_t} = 0, k \neq l.$$

### 3.3. The Closed form of $\widetilde{\mathscr{P}}_{\mathbb{T}_\mathcal{X}}$

With the parameterization and orthogonality between components in Lemma 3.2, we can derive the closed form of $\widetilde{\mathscr{P}}_{\mathbb{T}_\mathcal{X}}(\mathcal{A})$ for arbitrary tensor $\mathcal{A} \in \mathbb{R}^{n_1 \times n_2 \times n_3}$, denote

$$\widetilde{\mathscr{P}}_{\mathbb{T}_\mathcal{X}}(\mathcal{A}) := \mathcal{C}_\mathcal{A} \times_{i=1}^3 \widetilde{\boldsymbol{U}}_i + \sum_{i=1}^3 \mathcal{D} \times_i \boldsymbol{A}_i \times_{j \neq i} \widetilde{\boldsymbol{U}}_j$$

with unknown $\mathcal{C}_{\mathcal{A}} \in \mathbb{R}^{r_1 \times r_2 \times r_3}$ and $\widetilde{U}_i^\top G_{t,i} A_i = \mathbf{0}_{r_i}, i = 1, 2, 3$, then it is equivalent to solve the following problem

$$\min_{\mathcal{C}_{\mathcal{A}}, \{A_i\}_{i=1}^3} \left\| \mathcal{C}_{\mathcal{A}} \times_{i=1}^3 \widetilde{U}_i + \sum_{i=1}^3 \mathcal{D} \times_i A_i \times_{j \neq i} \widetilde{U}_j - \mathcal{A} \right\|_{\mathscr{W}_t}^2$$

$$s.t. \ \widetilde{U}_i^\top G_{t,i} A_i = \mathbf{0}_{r_i}, \ i = 1, 2, 3.$$

Here $\|\cdot\|_{\mathscr{W}_t}$ is the induced norm of inner product $\langle \cdot, \cdot \rangle_{\mathscr{W}_t}$. Due to Proposition 3.3, $\mathcal{C}_{\mathcal{A}}$ and $A_i$ can be solved individually:

$$\mathcal{C}_{\mathcal{A}} = \underset{\mathcal{C}_{\mathcal{A}}}{\arg\min} \left\| \mathcal{C}_{\mathcal{A}} \times_{i=1}^3 \widetilde{U}_i - \mathcal{A} \right\|_{\mathscr{W}_t}^2,$$

$$A_i = \underset{\widetilde{U}_i^\top G_{t,i} A_i = \mathbf{0}_{r_i}}{\arg\min} \left\| \mathcal{D} \times_i A_i \times_{j \neq i} \widetilde{U}_j - \mathcal{A} \right\|_{\mathscr{W}_t}^2.$$

Thus, the explicit form of $\widetilde{\mathscr{P}}_{\mathbb{T}_\mathcal{X}}$ can be derived, as summarized in the following lemma:

**Lemma 3.4.** *For tensor $\mathcal{X} \in \mathbb{M}_r$ and $\widehat{\mathcal{X}} = \mathscr{W}_t^{\frac{1}{2}} \mathcal{X} := \mathcal{X} \times_{i=1}^3 G_{t,i}^{\frac{1}{2}} \in \mathbb{M}_r$, denote $\mathscr{P}_{\mathbb{T}_{\widehat{\mathcal{X}}}}$ as projection operators onto tangent space $\mathbb{T}_{\widehat{\mathcal{X}}}$ of $\widehat{\mathcal{X}}$ under Euclidean metric, then:*

$$\widetilde{\mathscr{P}}_{\mathbb{T}_\mathcal{X}} = \mathscr{W}_t^{-\frac{1}{2}} \cdot \mathscr{P}_{\mathbb{T}_{\widehat{\mathcal{X}}}} \cdot \mathscr{W}_t^{\frac{1}{2}}.$$

Lemma 3.4 indicates that compared to RGD, computing the Riemannian gradient $\widetilde{\mathscr{P}}_{\mathbb{T}_{\mathcal{X}^t}} \mathscr{W}_t^{-1} \mathcal{G}^t$ in PRGD requires two additional operations. The first operation, $\mathscr{W}_t^{-\frac{1}{2}} \mathcal{G}^t$, costs $O(|\Omega|)$ since $\mathcal{G}^t$ is sparse. For the second operation $\mathscr{W}_t^{-\frac{1}{2}} \left( \mathscr{P}_{\mathbb{T}_{\widehat{\mathcal{X}}^t}} \mathscr{W}_t^{-\frac{1}{2}} \mathcal{G}^t \right)$, $\mathscr{P}_{\mathbb{T}_{\widehat{\mathcal{X}}^t}} \mathscr{W}_t^{-1} \mathcal{G}^t$ lies in $\mathbb{T}_{\widehat{\mathcal{X}}^t}$ thus can be expressed as the sum of four multilinear-rank-$r$ tensors. The computational cost is $O(nr)$. Consequently, our proposed preconditioned strategy is efficient, as the additional computational overhead is negligible compared to the computational cost of RGD, which scales as $O(n^3 r)$ (Cai et al., 2020).

### 3.4. PRGD Algorithm

The details of the proposed PRGD algorithm, based on the derivation of the Riemannian gradient under $\mathscr{W}_t$ above, are summarized in Algorithm 1.

---

**Algorithm 1** Preconditioned RGD

**Initialization:** $\mathcal{X}^0 \in \mathbb{M}_r$ and spikiness parameter $\nu$
**for** $t = 0, 1, \ldots, T_{\max}$ **do**
  $\mathcal{G}^t = \mathscr{P}_\Omega(\mathcal{X}^t) - \mathscr{P}_\Omega(\mathcal{X}_*)$.
  $\mathcal{W}^t = \mathcal{X}^t - \alpha_t \widetilde{\mathscr{P}}_{\mathbb{T}_{\mathcal{X}^t}} \mathscr{W}_t^{-1} \mathcal{G}^t$.
  $\overline{\mathcal{W}}^t = \mathrm{Trim}_{\xi_t}(\mathcal{W}^t)$ with $\xi_t = \frac{8\|\mathcal{W}^t\|_F}{7 n^{3/2} \nu}$.
  $\mathcal{X}^{t+1} = \mathscr{H}_r(\overline{\mathcal{W}}^t)$.
**end for**

---

Algorithm 1 involves an additional procedure called trimming, which is used to guarantee the incoherence property of iterates in matrix or tensor completion task (Wei et al., 2020; Tong et al., 2022; Cai et al., 2022b). The trimming operator $\mathrm{Trim}_\zeta$ is defined as follows, for given tensor $\mathcal{W}$,

$$[\mathrm{Trim}_\zeta(\mathcal{W})]_{i,j,k} = \begin{cases} \zeta \cdot \mathrm{Sign}(\mathcal{W}_{i,j,k}), & \text{if } |\mathcal{W}_{i,j,k}| \geq \zeta \\ \mathcal{W}_{i,j,k} & \text{otherwise.} \end{cases}$$

This operation trims large entries of $\mathcal{W}$, ensuring that its spikiness remains within an acceptable level. Consequently, it guarantees the incoherence property of the iterates by Lemma 2.2.

Theorem 4.1 demonstrates that, under appropriate initialization conditions, Algorithm 1 achieves linear convergence to the global minimizer. To satisfy these conditions, we propose an initialization algorithm presented in Algorithm 2. Leveraging the subspace estimators introduced in (Cai et al., 2021), Algorithm 2 first estimates the factor matrices $U_i$ of $\mathcal{W}^0$ using the top-$r$ eigenvectors of the diagonally-deleted Gram matrix of $p^{-1} \mathscr{M}_i(\mathscr{P}_\Omega(\mathcal{X}_*))$ (we use $\mathscr{P}_{\text{off-diag}} = \mathscr{I} - \mathscr{P}_{\text{diag}}$ and $\mathscr{I}$ is the identity operator). Subsequently, the algorithm constructs the initialization $\mathcal{X}^0$ by applying the trimming procedure on $\mathcal{W}^0$ to ensure the reconstructed tensor satisfies the required spikiness condition.

---

**Algorithm 2** Spectral Initialization with Trimming

**Input:** $\mathscr{P}_\Omega(\mathcal{X}^*) \in \mathbb{R}^{n_1 \times n_2 \times n_3}$, multilinear rank $r$, sampling ratio $p$ and spikiness parameter $\nu$.
**for** $i = 1, 2, 3$ **do**
  $T_i = p^{-1} \mathscr{M}_i(\mathscr{P}_\Omega(\mathcal{X}_*))$.
  $U_i \leftarrow$ the top-$r_i$ eigenvectors of $\mathscr{P}_{\text{off-diag}}(T_i T_i^\top)$.
**end for**
$\mathcal{W}^0 = p^{-1} \mathscr{P}_\Omega(\mathcal{X}_*) \times_{i=1}^3 U_i U_i^\top$.
**Output:** $\mathcal{X}^0 = \mathscr{H}_r(\mathrm{Trim}_\xi(\mathcal{W}^0))$ with $\xi = \frac{8\|\mathcal{W}^0\|_F}{7 n^{3/2} \nu}$.

---

Here we remark that the parameter $\nu$ in Algorithm 1 and Algorithm 2 is not necessarily the true spikiness condition. Instead, it can be treated as a tuning parameter set slightly larger than $Spiki(\mathcal{X}_*)$. Furthermore, we numerically observe that the algorithm performs nearly the same with and without the trimming procedure. Therefore, from a practical perspective, the trimming step can be omitted.

## 4. Recovery Guarantee

The convergence analysis of PRGD is divided into two parts, local linear convergence and initialization estimation. We first present the local linear convergence of PRGD in Theorem 4.1.

**Theorem 4.1.** *Suppose $\mathcal{X}_* \in \mathbb{M}_r$ with $Spiki(\mathcal{X}_*) \leq \nu$, $\Omega$ satisfies the Bernoulli model (i.e. each entry is observed*

*independently with probability p) and the initialization satisfies*

$$\|\mathcal{X}^0 - \mathcal{X}_*\|_F \leq \frac{\sigma_{\min}(\mathcal{X}_*)}{\widetilde{C} r^{\frac{1}{2}}} \quad and \quad Incoh(\mathcal{X}^0) \leq 2\nu\kappa_0$$

*for a sufficiently large but absolute constant $\widetilde{C} > 0$. Then there exist an absolute constant $C > 0$ such that if $p$ satisfies*

$$p \geq C \log^3(n)\bar{n}^{-\frac{3}{2}} r^{\frac{7}{2}} \nu^3 \kappa_0^5$$
$$+ C \left(\log^5(n) r^7 \nu^6 \kappa_0^6 \vee \log(n) r^8 \nu^2 \kappa_0^2\right) n\bar{n}^{-3}\kappa_0^4,$$

*then with high probability (i.e. at least $1 - c_1 n^{-c_2}$ for $c_1, c_2 > 0$ and $c_2$ sufficiently large), by choosing $\epsilon_t$ computed from the gradient and step size $\alpha_t = \epsilon_t^{\frac{1}{2}} p^{-1}$, the sequence $\{\mathcal{X}^t\}_{t=1}^{\infty}$ generated by Algorithm 1 satisfy*

$$\|\mathcal{X}^{t+1} - \mathcal{X}_*\|_F \leq 0.351 \cdot \|\mathcal{X}^t - \mathcal{X}_*\|_F,$$

*for all $t = 0, 1, 2\ldots$.*

Theorem 4.1 dictates that Algorithm 1 converges linearly to $\mathcal{X}^*$ provided that the initialization $\mathcal{X}^0$ is sufficiently close to $\mathcal{X}_*$ and the incoherence condition is satisfied. By leveraging the spectral initialization property from (Tong et al., 2022) and the trimming property from (Cai et al., 2023), we derive the following estimation for Algorithm 2, ensuring it meets the initialization error requirements specified in Theorem 4.1.

**Lemma 4.2.** *Assume the condition of $\mathcal{X}^*$ and $\Omega$ from Theorem 4.1 holds. Then there exists an absolute constant $C > 0$ such that if*

$$p \geq C \log^3(n)\bar{n}^{-\frac{3}{2}} r^3 \nu^3 \kappa_0^5 + C \log^5(n) n\bar{n}^{-3} r^5 \nu^4 \kappa_0^8,$$

*then with high probability, the initialization tensor $\mathcal{X}^0$ from Algorithm 2 satisfies*

$$\|\mathcal{X}^0 - \mathcal{X}_*\|_F \leq \frac{\sigma_{\min}(\mathcal{X}_*)}{\widetilde{C} r^{\frac{1}{2}}} \quad and \quad Incoh(\mathcal{X}^0) \leq 2\nu\kappa_0.$$

By directly combining Theorem 4.1 with Lemma 4.2, the recovery guarantee of the PRGD algorithm can be established with a nearly optimal sampling complexity $O(\bar{n}^{3/2})$. Detailed proofs of Theorem 4.1 and Lemma 4.2 are provided in the Appendix B.

# 5. Numerical Experiments

In this section, we proposed several numerical comparisons of our proposed PRGD algorithm with state-of-the-art algorithms that include RGD (Cai et al., 2020; Wang et al., 2023), for demonstrating the effectiveness of precondition, and ScaledGD (Tong et al., 2022), for comparisons with factorization based algorithms. For our PRGD algorithm, we keep the hyperparamter $\epsilon_t$ a constant chosen from

$\{10^{-3}, 5 \times 10^{-4}, 10^{-4}, 5 \times 10^{-5}\}$. Since no step size strategy is proposed for the ScaledGD algorithm in (Tong et al., 2022), to make a fair comparison, we tune the optimal constant step size for all tested algorithms and report the results. All simulations are performed in MATLAB r2023b with a 2.6GHZ Intel Xeon ICX Platinum 8358 CPU.

We test those algorithms on both synthetic data and real data. For the synthetic data, we set $n_1 = n_2 = n_3 = n$ and $r_1 = r_2 = r_3 = r$. Following the same manner in (Kressner et al., 2014), we construct the ground truth tensor $\mathcal{X}_*$ by first generating the entries of core tensor and factor matrices from a uniform distribution on $[0, 1]$, then use QR decomposition on the random factor matrices to obtain the orthonormal ones. For the real data, we consider the video inpainting problem, where the goal is to reconstruct the original video from its partially observed pixels. Throughout this section, we use the relative error frequently that is defined by

$$\text{relative error} = \frac{\|\mathcal{X}^{\text{out}} - \mathcal{X}_*\|_F}{\|\mathcal{X}_*\|_F},$$

where $\mathcal{X}^{\text{out}}$ is the output tensor of the algorithm.

## 5.1. Reocovery Ability

**Phase Transition.** We first explore the recovery abilities of PRGD in the framework of phase transition. A test is considered to be successful if the recovered tensor has a relative error smaller than $10^{-3}$. For each set of parameters, we run 100 random trails and count the success rate. The results in Figure 1 demonstrate the recovery is successful when the sampling size is moderately large. And when the rank increases, more sampling is needed for a successful recovery. These are consistent with our theoretical findings.

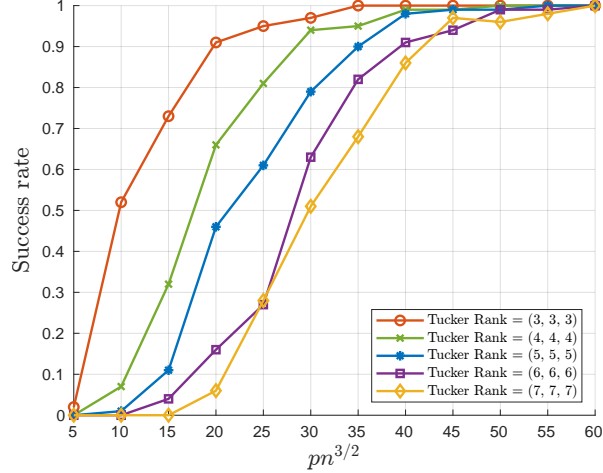

*Figure 1.* The success rate of PRGD with respect to the sample size $pn^{3/2}$ for tensor with size $n = 100$ and rank $r$ ranging from 3 to 7.

### 5.2. Comparison with Other Algorithms

**Iteration Count and Runtime.** We next investigate the iteration count and runtime of all tested algorithms under different oversampling ratios (OS), which is defined as

$$\text{OS} = \frac{|\Omega|}{\dim(\mathbb{M}_{\boldsymbol{r}})} = \frac{pn^3}{r^3 + 3(nr - r^2)}.$$

Here we consider the tensors of size $n = 100, 150, 200$ and rank $r = 5$. The OSs are set to $10, 15, 20$ to represent different sampling levels. We count the iteration numbers and runtimes of the tested algorithm until the relative error is less than $10^{-4}$. For each parameter setting, we perform five random trials and report the average results in Figure 2 and Figure 3.

The results in Figure 2 demonstrate that our PRGD algorithm significantly outperforms the other two algorithms in terms of speed under various parameter settings, achieving approximately a $10\times$ speedup compared to RGD and a 3 to $4\times$ speedup compared to ScaledGD. Additionally, the results in Figure 3 are consistent with those in Figure 2, further verifying that the additional computational cost per iteration of PRGD is negligible compared to RGD.

**Increase the Sampling Level.** To highlight the advantage of using preconditioning across different sampling levels. We fix the tensor size $n = 100$ and rank $r = 5$ while increasing the OS from 10 to 100, corresponding to sampling ratio $p$ from 1.6% to 16%. We perform five random trials for each OS and report the average results in Figure 4.

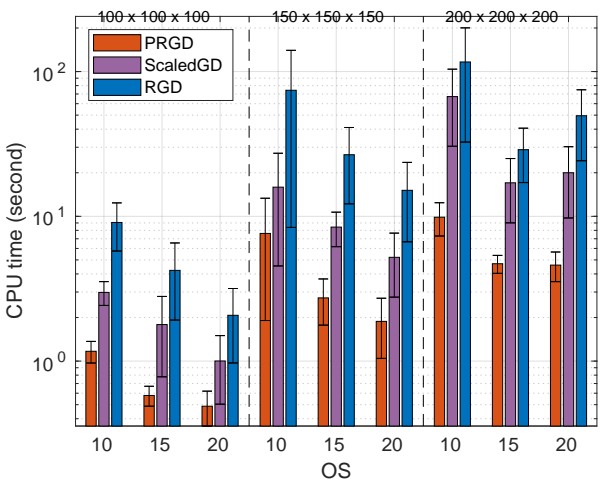

*Figure 2.* The runtime results of tested algorithms in different settings.

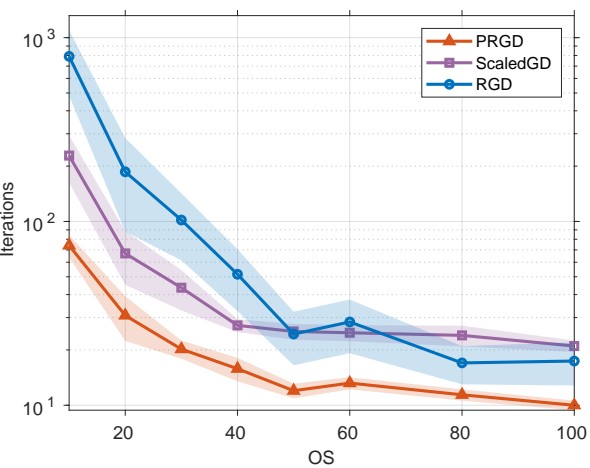

*Figure 4.* The iteration counts of three algorithms at different sampling levels.

Figure 4 shows that when the sampling level is low, the benefits of precondition are substantial compared to RGD. As the sampling level increases, RGD gradually improves, outperforming ScaledGD and approaching the performance of PRGD.

**Noisy Data Reconstruction.** We then investigate the property of the PRGD algorithm in the presence of noise. For ground truth tensor $\mathcal{X}_*$, the known entries of $\mathcal{X}_*$ are perturbed by rescaled Gaussian noise $\mathcal{E}$, such that $\|\mathscr{P}_\Omega \mathcal{E}\|_F = \sigma \cdot \|\mathscr{P}_\Omega \mathcal{X}_*\|_F$ for a given noise level $\sigma$. We fix the size $n = 100$, rank $r = 10$, OS $= 10$ and noise level $\sigma$ from $5 \times 10^{-4}$ to $10^{-1}$. The stopping criteria is chosen when $\|\mathcal{X}^{t+1} - \mathcal{X}^t\|_F / \|\mathcal{X}^t\|_F \leq 10^{-4}$. For each parameter setting, we repeat five random trails and we report the average relative error when the stopping criteria is satisfied. The results shown in Figure 5 and Appendix C indicate our PRGD is faster than others and robust to the additive noise.

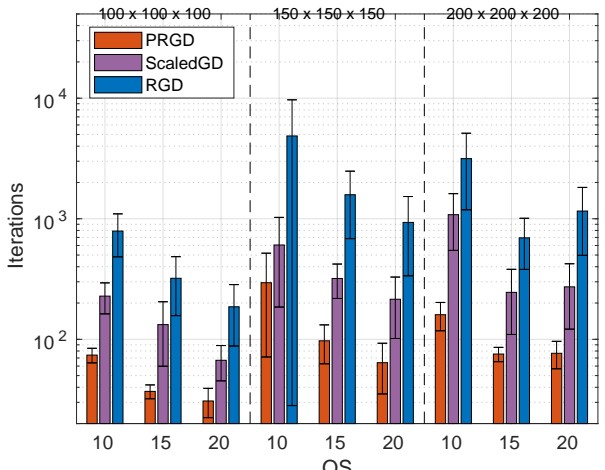

*Figure 3.* The iteration numbers of tested algorithms in different settings.

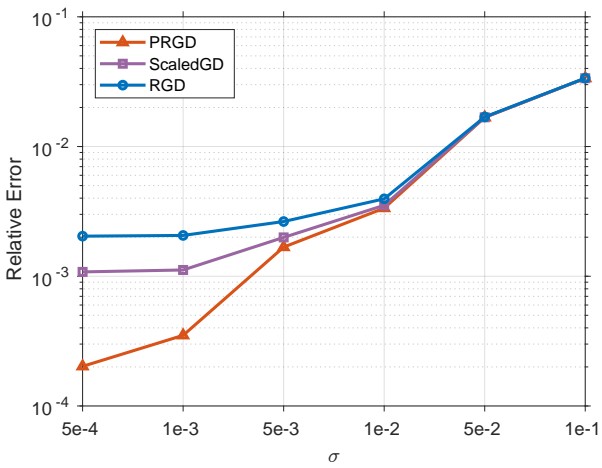

*Figure 5.* Noisy data reconstruction results at different noise levels.

### 5.3. Real Data Reconstruction

**Video Inpainting.** We evaluate the performance of our proposed algorithm on the *Tomato* video from (Liu et al., 2012), as well as the *Akiyo* and *Hall Monitor* videos from the YUV video dataset[1]. For the *Akiyo* and *Hall Monitor* videos, only the first 100 frames are used. Since these videos are in RGB format, we concatenate their three colour channels along the third dimension and treat each video as a three-order tensor.

*Table 1.* PSNR values of the outputs generated by different algorithms for three test videos with different multilinear ranks.

| VIDEO(SIZE) | $r$ | PRGD | RGD | SCALEDGD |
|---|---|---|---|---|
| TOMATO | 50 | **27.34** | 27.15 | 27.09 |
| | 60 | **27.79** | 27.44 | 27.15 |
| (242, 320, 501) | 70 | **28.25** | 27.50 | 27.15 |
| HALL MONITOR | 50 | **27.61** | 27.43 | 26.68 |
| | 60 | **28.34** | 28.06 | 26.25 |
| (288, 352, 300) | 70 | **28.83** | 28.75 | 28.52 |
| AKIYO | 50 | **29.29** | 29.09 | 28.81 |
| | 60 | **29.95** | 29.69 | 28.67 |
| (288, 352, 300) | 70 | **30.45** | 30.42 | 28.62 |

We assume that only 10% of the pixels are observed for each video and employ different algorithms to reconstruct the original video. We vary the multilinear rank $(r, r, r)$ and report the Peak Signal-to-Noise Ratio (PSNR) of the reconstructed videos in Table 1. The image of certain frames of reconstructed videos and more detailed results are available in Appendix. C. It can be observed that our PRGD algorithm consistently achieves higher PSNR values compared to other algorithms across different settings and datasets, demonstrating its effectiveness for video inpainting prob-

---

[1]http://trace.eas.asu.edu/yuv/

lems.

## 6. Conclusion

In this work, we propose a preconditioned Riemannian gradient descent algorithm for the low-multilinear-rank tensor completion problem. The preconditioned metric is data-driven and computationally efficient. Numerical experiments demonstrate that the proposed PRGD achieves approximately $10\times$ acceleration compared to the standard RGD. Furthermore, we establish the linear convergence of PRGD and provide a recovery guarantee with near-optimal sampling complexity. Lastly, we summarize several promising directions for future research.

- *Other low-multilinear-rank applications.* Practical data often contain noise and outliers. In this work, we evaluate PRGD numerically on noisy data. However, a theoretical analysis of robust tensor completion and tensor regression remains for further investigation.

- *Entrywise convergence analysis of PRGD.* In this work, our convergence analysis focuses on estimating the $\ell_2$ error of iterates. In contrast, (Wang et al., 2023) directly deals with $\ell_\infty$ error, thus they can avoid the trimming operations and simultaneously maintain the incoherence condition of iterates below an acceptable level. It would be interesting to investigate whether PRGD can exhibit similar properties.

- *Other preconditioned Riemannian optimization methods.* Building upon the preconditioned Riemannian metric proposed in this work, it is worthwhile to investigate extensions such as the preconditioned Riemannian conjugate gradient descent and Gauss-Newton methods. Furthermore, exploring new preconditioned Riemannian metrics is a promising direction for future research. By leveraging the tangent space parameterization framework in this study, the corresponding Riemannian optimization methods can be readily derived.

## Acknowledgements

The authors would like to thank the four anonymous reviewers for their valuable comments. The work of Jian-Feng Cai was supported by the Hong Kong Research Grants Council GRF Grants 16307023 and 16306124. Xiaoqun Zhang was supported by the National Science Foundation of China (No.120990024) and Shanghai Municipal Science and Technology Major Project (2021SHZDZX0102).

## Impact Statement

This paper presents work that aims to advance the field of tensor-related optimization without any known detrimental broader impacts. The insights and methods proposed in this study are intended to make a positive contribution to tensor-related optimization and related areas.

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

# A. Appendix

### Proof of Proposition 3.1

*Proof.* For $\mathcal{X} \in \mathbb{M}_r$, we derive the Riemannian gradient $\widetilde{\operatorname{grad}} f(\mathcal{X})$ under the data-driven metric $\langle \cdot, \cdot \rangle_{\mathscr{W}_t}$. Let $\gamma(s)$ be a smooth curve in $\mathbb{M}_r$ with $\gamma(0) = \mathcal{X}$. Denote $\widetilde{\mathscr{P}}_{\mathbb{T}_{\mathcal{X}}}$ be the orthogonal projection operator onto $\mathbb{T}_{\mathcal{X}}$ under $\langle \cdot, \cdot \rangle_{\mathscr{W}_t}$ and $\mathcal{G}$ is the gradient on $\mathcal{X}$, then

$$\frac{d}{ds} f(\gamma(s))\big|_{s=0} = \langle \dot{\gamma}(0), \mathcal{G} \rangle = \langle \dot{\gamma}(0), \widetilde{\mathscr{P}}_{\mathbb{T}_{\mathcal{X}}} \mathscr{W}_t^{-1} \mathcal{G} \rangle_{\mathscr{W}_t},$$

therefore, the Riemannian gradient is

$$\widetilde{\operatorname{grad}} f(\mathcal{X}) = \widetilde{\mathscr{P}}_{\mathbb{T}_{\mathcal{X}}} \mathscr{W}_t^{-1} \mathcal{G} \in \mathbb{T}_{\mathcal{X}}.$$

$\square$

### Proof of Lemma 3.2

*Proof.* For a symmetric positive definite matrix $\boldsymbol{G} \in \mathbb{R}^{n \times n}$, $\operatorname{St}(n, r, \boldsymbol{G}) := \{\boldsymbol{U} \in \mathbb{R}^{n \times r} | \boldsymbol{U}^\top \boldsymbol{G} \boldsymbol{U} = \boldsymbol{I}_r\}$ is the generized Stiefel manifold. The tangent space at $\boldsymbol{U} \in \operatorname{St}(n, r, \boldsymbol{G})$ is

$$\mathcal{T}_{\boldsymbol{U}} \operatorname{St}(n, r, \boldsymbol{G}) := \{\dot{\boldsymbol{U}} \in \mathbb{R}^{n \times r} : \dot{\boldsymbol{U}}^\top \boldsymbol{G} \boldsymbol{U} + \boldsymbol{U}^\top \boldsymbol{G} \dot{\boldsymbol{U}} = \boldsymbol{0}_r\} = \{\dot{\boldsymbol{U}} \in \mathbb{R}^{n \times r} : \boldsymbol{U}^\top \boldsymbol{G} \dot{\boldsymbol{U}} \in \mathbb{S}_{\text{skew}}(r)\},$$

where $\mathbb{S}_{\text{skew}}(r)$ denotes the space of skew-symmetric real $r \times r$ matrices. Consider the extended tangent map of $(\mathcal{S}, \boldsymbol{U}_1, \boldsymbol{U}_2, \boldsymbol{U}_3) \mapsto \mathcal{Y} = \mathcal{S} \times_{i=1}^3 \boldsymbol{U}_i$,

$$\mathbb{R}^{r \times r \times r} \times \prod_{i=1}^3 \mathcal{T}_{\boldsymbol{U}} \operatorname{St}(n, r, \boldsymbol{G}_i) \to \mathbb{T}_{\mathcal{Y}} \times \prod_{i=1}^3 \mathbb{S}_{\text{skew}}(r),$$

$$\left(\dot{\mathcal{S}}, \dot{\boldsymbol{U}}_1, \dot{\boldsymbol{U}}_2, \dot{\boldsymbol{U}}_3\right) \mapsto \left(\dot{\mathcal{S}} \times_{i=1}^3 \boldsymbol{U}_i + \sum_{i=1}^3 \mathcal{S} \times_i \dot{\boldsymbol{U}}_i \times_{k \neq i} \boldsymbol{U}_k, \boldsymbol{U}_1^T \boldsymbol{G}_1 \dot{\boldsymbol{U}}_1, \boldsymbol{U}_2^T \boldsymbol{G}_2 \dot{\boldsymbol{U}}_2, \boldsymbol{U}_3^T \boldsymbol{G}_3 \dot{\boldsymbol{U}}_3\right),$$

where $\dot{\mathcal{S}} \in \mathbb{R}^{r_1 \times r_2 \times r_3}$ and $\dot{\boldsymbol{U}}_i \in \mathcal{T}_{\boldsymbol{U}_i} \operatorname{St}(n_i, r_i, \boldsymbol{G}_i)$. Moreover, for $\dot{\mathcal{Y}} \in \mathbb{T}_{\mathcal{Y}}$, the $\dot{\mathcal{S}}$ and $\dot{\boldsymbol{U}}_i$ are unique determined if we impose

$$\boldsymbol{U}_i^\top \boldsymbol{G}_i \dot{\boldsymbol{U}}_i = \boldsymbol{0}_{r_i}, i = 1, 2, 3.$$

Then $\dot{\mathcal{S}}$ and $\dot{\boldsymbol{U}}_i$ are given by the following formulas:

$$\begin{aligned}
\dot{\mathcal{S}} &= \dot{\mathcal{Y}} \times_{i=1}^3 (\boldsymbol{U}_i^\top \boldsymbol{G}_i), \\
\dot{\boldsymbol{U}}_i &= (\boldsymbol{I}_{n_i} - \boldsymbol{U}_i \boldsymbol{U}_i^\top \boldsymbol{G}_i) \mathscr{M}_i \left(\dot{\mathcal{Y}} \times_{k \neq i} (\boldsymbol{U}_k^\top \boldsymbol{G}_k)\right) \mathscr{M}_i(\mathcal{S})^\dagger, i = 1, 2, 3.
\end{aligned}$$

Thus the extended tangent map is an isomorphism and we complete the proof. $\square$

### Proof of Proposition 3.3

*Proof.* Note that for $\mathcal{Z} = \mathcal{S} \times_{i=1}^3 \widehat{\boldsymbol{U}}_i$, the matricization of $\mathcal{Z}$ satisfies

$$\begin{aligned}
\mathscr{M}_1(\mathcal{Z}) &= \widehat{\boldsymbol{U}}_1 \mathscr{M}_1(\mathcal{S}) \boldsymbol{M}_1^\top, \quad \boldsymbol{M}_1 = \widehat{\boldsymbol{U}}_3 \otimes \widehat{\boldsymbol{U}}_2 \\
\mathscr{M}_2(\mathcal{Z}) &= \widehat{\boldsymbol{U}}_2 \mathscr{M}_2(\mathcal{S}) \boldsymbol{M}_2^\top, \quad \boldsymbol{M}_2 = \widehat{\boldsymbol{U}}_3 \otimes \widehat{\boldsymbol{U}}_1 \\
\mathscr{M}_3(\mathcal{Z}) &= \widehat{\boldsymbol{U}}_3 \mathscr{M}_3(\mathcal{S}) \boldsymbol{M}_3^\top, \quad \boldsymbol{M}_3 = \widehat{\boldsymbol{U}}_2 \otimes \widehat{\boldsymbol{U}}_1.
\end{aligned}$$

Denote $\widehat{\boldsymbol{U}}_i := \boldsymbol{G}_{t,i}^{\frac{1}{2}} \widetilde{\boldsymbol{U}}_i, \widehat{\boldsymbol{W}}_i := \boldsymbol{G}_{t,i}^{\frac{1}{2}} \widetilde{\boldsymbol{W}}_i, i = 1, 2, 3$, then

$$\begin{aligned}
\langle \mathcal{C} \times_{i=1}^3 \widetilde{\boldsymbol{U}}_i, \mathcal{D} \times_i \widetilde{\boldsymbol{W}}_i \times_{j \neq i} \widetilde{\boldsymbol{U}}_j \rangle_{\mathscr{W}_t} &= \langle \mathcal{C} \times_{i=1}^3 \widehat{\boldsymbol{U}}_i, \mathcal{D} \times_i \widehat{\boldsymbol{W}}_i \times_{j \neq i} \widehat{\boldsymbol{U}}_j \rangle \\
&= \langle \widehat{\boldsymbol{U}}_i \mathscr{M}_i(\mathcal{C}) \boldsymbol{M}_i^\top, \widehat{\boldsymbol{W}}_i \mathscr{M}_i(\mathcal{D}) \boldsymbol{M}_i^\top \rangle \\
&= 0
\end{aligned}$$

where the third equality is due to $\widehat{\boldsymbol{U}}_k^\top \widehat{\boldsymbol{W}}_k = \widetilde{\boldsymbol{U}}_k^\top \boldsymbol{G}_{t,i} \widehat{\boldsymbol{W}}_k = \boldsymbol{0}_r$. Similarly, when $k \neq l$

$$\langle \mathcal{D} \times_k \widehat{\boldsymbol{W}}_k \times_{j \neq k} \widetilde{\boldsymbol{U}}_j, \mathcal{D} \times_l \widehat{\boldsymbol{W}}_l \times_{j \neq l} \widetilde{\boldsymbol{U}}_j \rangle_{\mathscr{W}_t} = \langle \mathcal{D} \times_k \widehat{\boldsymbol{W}}_k \times_{j \neq k} \widehat{\boldsymbol{U}}_j, \mathcal{D} \times_l \widehat{\boldsymbol{W}}_l \times_{j \neq l} \widehat{\boldsymbol{U}}_j \rangle$$
$$= \langle \widehat{\boldsymbol{W}}_k \mathscr{M}_k(\mathcal{D}) \boldsymbol{M}_k^\top, \widehat{\boldsymbol{U}}_k \mathscr{M}_k(\mathcal{D}) \boldsymbol{M}_{k,l}^\top \rangle$$
$$= 0$$

here for simplicity, we denote $\mathscr{M}_k(\mathcal{D} \times_l \widehat{\boldsymbol{W}}_l \times_{j \neq l} \widehat{\boldsymbol{U}}_j) = \widehat{\boldsymbol{U}}_k \mathscr{M}_k(\mathcal{D}) \boldsymbol{M}_{k,l}^\top$ for some matrix $\boldsymbol{M}_{k,l}$. $\qquad \square$

**Proof of Lemma 3.4**

*Proof.* For $\mathcal{X} \in \mathbb{M}_r$, we denote $\widehat{\mathcal{X}} := \mathscr{W}_t^{\frac{1}{2}}(\mathcal{X})$ then $\widehat{\mathcal{X}} \in \mathbb{M}_r$ since the $\mathscr{W}_t$ operation doesn't changes the multilinear rank. We denote $\widehat{\mathcal{X}} = \mathcal{D} \times_{i=1}^3 \widehat{\boldsymbol{U}}_i$ with $\widehat{\boldsymbol{U}}_i^\top \widehat{\boldsymbol{U}}_i = \boldsymbol{I}_{r_i}, i = 1, 2, 3$ as the standard Tucker decomposition. Let $\widetilde{\boldsymbol{U}}_i = \boldsymbol{G}_{t,i}^{-\frac{1}{2}} \widehat{\boldsymbol{U}}_i, i = 1, 2, 3$, then $\mathcal{X} = \mathscr{W}_t^{-\frac{1}{2}}(\widehat{\mathcal{X}}) = \mathcal{D} \times_{i=1}^3 \widetilde{\boldsymbol{U}}_i$ with $\widetilde{\boldsymbol{U}}_i^\top \boldsymbol{G}_{t,i} \widetilde{\boldsymbol{U}}_i = \boldsymbol{I}_{r_i}, i = 1, 2, 3$.

By Lemma 3.2, for arbitrary tensor $\mathcal{A} \in \mathbb{R}^{n_1 \times n_2 \times n_3}$, the projection $\widetilde{\mathscr{P}}_{\mathbb{T}_\mathcal{X}}(\mathcal{A}) := \mathcal{C}_\mathcal{A} \times_{i=1}^3 \widetilde{\boldsymbol{U}}_i + \sum_{i=1}^3 \mathcal{D} \times_i \boldsymbol{A}_i \times_{j \neq i} \widetilde{\boldsymbol{U}}_j$, where $\mathcal{C}_\mathcal{A}$ and $\boldsymbol{A}_i$'s can be solved as follows:

$$\mathcal{C}_\mathcal{A} = \underset{\mathcal{C}_\mathcal{A}}{\operatorname{argmin}} \left\| \mathcal{C}_\mathcal{A} \times_{i=1}^3 \widetilde{\boldsymbol{U}}_i - \mathcal{A} \right\|_{\mathscr{W}_t}^2$$
$$= \underset{\mathcal{C}_\mathcal{A}}{\operatorname{argmin}} \left\| \mathcal{C}_\mathcal{A} \times_{i=1}^3 \widehat{\boldsymbol{U}}_i - \mathscr{W}_t^{\frac{1}{2}}(\mathcal{A}) \right\|_F^2$$
$$= \left( \mathscr{W}_t^{\frac{1}{2}}(\mathcal{A}) \right) \times_{i=1}^3 \widehat{\boldsymbol{U}}_i^\top.$$

where the third equality is due to $\widehat{\boldsymbol{U}}_i^\top \widehat{\boldsymbol{U}}_i = \boldsymbol{I}_{r_i}, i = 1, 2, 3$. While for each $\boldsymbol{A}_i$, denote $\widehat{\boldsymbol{A}}_i = \boldsymbol{G}_{t,i}^{\frac{1}{2}}(\boldsymbol{A}_i)$:

$$\boldsymbol{A}_i = \underset{\widetilde{\boldsymbol{U}}_i^\top \boldsymbol{G}_{t,i} \boldsymbol{A}_i = \boldsymbol{0}_{r_i}}{\operatorname{argmin}} \left\| \mathcal{D} \times_i \boldsymbol{A}_i \times_{j \neq i} \widetilde{\boldsymbol{U}}_j - \mathcal{A} \right\|_{\mathscr{W}_t}^2$$

thus

$$\widehat{\boldsymbol{A}}_i = \underset{\widehat{\boldsymbol{U}}_i^\top \widehat{\boldsymbol{A}}_i = \boldsymbol{0}_{r_i}}{\operatorname{argmin}} \left\| \mathcal{D} \times_i \widehat{\boldsymbol{A}}_i \times_{j \neq i} \widehat{\boldsymbol{U}}_j - \mathscr{W}_t^{\frac{1}{2}}(\mathcal{A}) \right\|_F^2$$
$$= \underset{\widehat{\boldsymbol{U}}_i^\top \widehat{\boldsymbol{A}}_i = \boldsymbol{0}_{r_i}}{\operatorname{argmin}} \left\| \widehat{\boldsymbol{A}}_i \mathscr{M}_i(\mathcal{D}) - \mathscr{M}_i \left( \mathscr{W}_t^{\frac{1}{2}}(\mathcal{A}) \times_{j \neq i} \widehat{\boldsymbol{U}}_j^\top \right) \right\|_F^2$$
$$= \left( \boldsymbol{I}_{n_i} - \widehat{\boldsymbol{U}}_i \widehat{\boldsymbol{U}}_i^\top \right) \mathscr{M}_i \left( \mathscr{W}_t^{\frac{1}{2}}(\mathcal{A}) \times_{j \neq i} \widehat{\boldsymbol{U}}_j^\top \right) \mathscr{M}_i^\dagger(\mathcal{D}).$$

where $\mathscr{M}_i^\dagger(\mathcal{D})$ is the Moore-Penrose pseudoinverse of $\mathscr{M}_i(\mathcal{D})$. Thus the components in $\widetilde{\mathscr{P}}_{\mathbb{T}_\mathcal{X}}(\mathcal{A})$ are equalivent to:

$$\mathcal{C}_\mathcal{A} \times_{i=1}^3 \widetilde{\boldsymbol{U}}_i = \mathscr{W}_t^{\frac{1}{2}}(\mathcal{A}) \times_{i=1}^3 \widehat{\boldsymbol{U}}_i^\top \times_{i=1}^3 \widetilde{\boldsymbol{U}}_i = \mathscr{W}_t^{-\frac{1}{2}} \left( \mathscr{W}_t^{\frac{1}{2}}(\mathcal{A}) \times_{i=1}^3 \widehat{\boldsymbol{U}}_i \widehat{\boldsymbol{U}}_i^\top \right)$$
$$\mathcal{D} \times_i \boldsymbol{A}_i \times_{j \neq i} \widetilde{\boldsymbol{U}}_j = \mathscr{W}_t^{-\frac{1}{2}} \left( \mathcal{D} \times_i \widehat{\boldsymbol{A}}_i \times_{j \neq i} \widehat{\boldsymbol{U}}_j \right), i = 1, 2, 3.$$

To conclude

$$\widetilde{\mathscr{P}}_{\mathbb{T}_\mathcal{X}}(\mathcal{A}) = \mathscr{W}_t^{-\frac{1}{2}} \cdot \mathscr{P}_{\mathbb{T}_{\widehat{\mathcal{X}}}} \cdot \mathscr{W}_t^{\frac{1}{2}}(\mathcal{A}).$$

As $\mathcal{A}$ is an arbitrary tensor, we complete the proof. $\qquad \square$

## B. Appendix

We first refer to the following lemmas:

**Lemma B.1** (Lemma 13.6.(Cai et al., 2023)). *Let $\mathcal{X}_* \in \mathbb{M}_r$ with spikiness parameter $\nu$. Assume $\mathcal{W}$ satisfies $\|\mathcal{W} - \mathcal{X}_*\|_F \leq \frac{\sigma_{\min}(\mathcal{X}_*)}{8}$. Then, the tensor $\mathscr{H}_r(\mathrm{Trim}_\zeta(\mathcal{W}))$ is incoherent with parameter $2\nu\kappa_0$ if we choose $\zeta = \frac{8\|\mathcal{W}\|_F}{7n^{3/2}}\nu$. Also, it satisfies:*

$$\|\mathscr{H}_r(\mathrm{Trim}_\zeta(\mathcal{W})) - \mathcal{X}_*\|_F \leq \|\mathcal{W} - \mathcal{X}_*\|_F + \frac{C_0 r^{1/2}\|\mathcal{W} - \mathcal{X}_*\|_F^2}{\sigma_{\min}(\mathcal{X}_*)}, \tag{6}$$

*where $C_0 > 0$ is a constant.*

**Lemma B.2** (Lemma 6. (Tong et al., 2022)). *Suppose that $\mathcal{X}_*$ is $\mu$-incoherent, $n_k \gtrsim \epsilon^{-1}\mu r_k^{\frac{3}{2}}\kappa_0$, and $p$ satisfies:*

$$p \gtrsim \epsilon^{-1}\bar{n}^{-\frac{3}{2}}\mu^3 r^{\frac{5}{2}}\kappa_0^2 \log^3(n) + \epsilon^{-2}n\bar{n}^{-3}\mu^4 r^4 \kappa_0^4 \log^5(n),$$

*for some small constant $\epsilon > 0$. Then with high probability, the spectral initialization $\mathcal{W}^0$ in Algorithm 2 satisfies:*

$$\|\mathcal{W}^0 - \mathcal{X}_*\|_F \leq \epsilon\sigma_{\min}(\mathcal{X}_*).$$

**Lemma B.3** (Lemma 20. (Tong et al., 2022)). *Suppose $\Omega$ satisfies the Bernoulli observation model. Then, with high probability,*

$$\left|\langle(\mathscr{I} - p^{-1}\mathscr{P}_\Omega)\mathcal{X}, \mathcal{Y}\rangle\right| \leq C_1\left(p^{-1}\log^3(n) + \sqrt{p^{-1}n\log^5(n)}\right)\tau,$$

*holds simultaneously for all $\mathcal{X}, \mathcal{Y} \in \mathbb{M}_r$, here $C_1$ is a constant. Denote $\mathcal{X} = \mathcal{C}_{\mathbf{X}} \times_{i=1}^3 \mathbf{X}_i, \mathcal{Y} = \mathcal{C}_{\mathbf{Y}} \times_{i=1}^3 \mathbf{Y}_i$, then the quantity $\tau$ obeys*

$$\tau \leq (\|\mathbf{X}_1\mathscr{M}_1(\mathcal{C}_{\mathbf{X}})\|_{2,\infty}\|\mathbf{Y}_1\mathscr{M}_1(\mathcal{C}_{\mathbf{Y}})\|_F \wedge \|\mathbf{X}_1\mathscr{M}_1(\mathcal{C}_{\mathbf{X}})\|_F\|\mathbf{Y}_1\mathscr{M}_1(\mathcal{C}_{\mathbf{Y}})\|_{2,\infty})$$
$$(\|\mathbf{X}_2\|_{2,\infty}\|\mathbf{Y}_2\|_F \wedge \|\mathbf{X}_2\|_F\|\mathbf{Y}_2\|_{2,\infty})(\|\mathbf{X}_3\|_{2,\infty}\|\mathbf{Y}_3\|_F \wedge \|\mathbf{X}_3\|_F\|\mathbf{Y}_3\|_{2,\infty}).$$

**Lemma B.4** (Lemma 18. (Tong et al., 2022)). *Suppose that $\mathcal{Z} = \mathcal{S} \times_{i=1}^3 \mathbf{V}_i \in \mathbb{M}_r$ is $\mu$-incoherent, and $\bar{n}^3 p \gtrsim \mu^4 nr^2 \log n$. Then, with high probability, one has:*

$$\left|\langle(\mathscr{I} - p^{-1}\mathscr{P}_\Omega)(\mathcal{X}_A), \mathcal{X}_B\rangle\right| \leq C_2\sqrt{\frac{\mu^4 nr^2 \log n}{p\bar{n}^3}}\|\mathcal{X}_A\|_F\|\mathcal{X}_B\|_F,$$

*simultaneously for all tensors $\mathcal{X}_A, \mathcal{X}_B \in \mathbb{R}^{n_1 \times n_2 \times n_3}$ in the form of*

$$\mathcal{X}_A = \sum_{i=1}^3 \mathcal{D}_{A,i} \times_{j\neq i} \mathbf{V}_j \times_i \mathbf{U}_{A,i}, \quad \mathcal{X}_B = \sum_{i=1}^3 \mathcal{D}_{B,i} \times_{j\neq i} \mathbf{V}_j \times_i \mathbf{U}_{B,i}$$

*where $\mathbf{U}_{A,i}, \mathbf{U}_{B,i} \in \mathbb{R}^{n\times r}, \mathcal{D}_{A,i}, \mathcal{D}_{B,i} \in \mathbb{R}^{r\times r\times r}, i = 1, 2, 3$ are arbitrary factors, and $C_2 > 0$ is a universal constant.*

**Lemma B.5** (Lemma 5.2. (Cai et al., 2020)). *Let $\mathcal{X}, \mathcal{X}_* \in \mathbb{M}_r$ be two order-3 multilinear-rank-$r$ tensors, then we have*

$$\|(\mathscr{I} - \mathscr{P}_{\mathbb{T}_\mathcal{X}})\mathcal{X}_*\|_F \leq \frac{7}{\sigma_{\min}(\mathcal{X}_*)}\|\mathcal{X} - \mathcal{X}_*\|_F^2. \tag{7}$$

**Lemma B.6** (Lemma 4.5. (Wei et al., 2020)). *Let $\mathbf{U}_1, \mathbf{U}_2 \in \mathbb{R}^{n\times r}$ be two orthogonal matrices. Then there exists a $r \times r$ unitary matrix $\mathbf{Q}$ such that*

$$\|\mathbf{U}_1 - \mathbf{U}_2\mathbf{Q}\|_F \leq \|\mathbf{U}_1\mathbf{U}_1^\top - \mathbf{U}_2\mathbf{U}_2^\top\|_F.$$

**Lemma B.7.** *For any $\mathcal{Z} \in \mathbb{R}^{n_1 \times n_2 \times n_3}$ and $t \in \mathbb{N}$, we have*

$$\nu_t\|\mathcal{Z}\|_F^2 \leq \|\mathcal{Z}\|_{\mathscr{W}_t}^2 \leq \mu_t\|\mathcal{Z}\|_F^2, \tag{8}$$

*where $\nu_t = \epsilon_t^{\frac{1}{2}}$ and $\mu_t = (\epsilon_t + \|\mathcal{G}_t\|_\vee^2)^{\frac{1}{2}}$. Here we define $\|\mathcal{G}_t\|_\vee := \max_{i=1,2,3}\max_k \|\mathscr{M}_i(\mathcal{G}_t)(k,:)\|_2$ be the maximum of $\mathcal{G}_t$'s slice norms and $\rho_t = \mu_t/\nu_t$.*

**Lemma B.8.** *The operator* $(\mathscr{W}_t^{-1} - \epsilon_t^{-\frac{1}{2}}\mathscr{I})$ *have the following bound:*

$$\left\|\mathscr{W}_t^{-1} - \epsilon_t^{-\frac{1}{2}}\mathscr{I}\right\| \leq \epsilon_t^{-\frac{1}{2}}\left(1 - \left(1 + \frac{\|\mathcal{G}_t\|_{\vee}^2}{\epsilon_t}\right)^{-\frac{1}{2}}\right). \tag{9}$$

*Proof.* It is straightforward since $(\epsilon_t + \|\mathcal{G}_t\|_{\vee}^2)^{-\frac{1}{2}} \leq \|\mathscr{W}_t^{-1}\| \leq \epsilon_t^{-\frac{1}{2}}$. $\qquad\square$

**Lemma B.9.** *For tensor* $\mathcal{X} \in \mathbb{M}_{\boldsymbol{r}}$ *is* $\mu$*-incoherent, then* $\widehat{\mathcal{X}} := \mathscr{W}_t^{\frac{1}{2}}\mathcal{X}$ *is* $(\rho_t^{\frac{1}{3}}\mu)$*-incoherent.*

*Proof.* Suppose $\mathcal{X} = \mathcal{D} \times_{i=1}^3 \boldsymbol{U}_i$ with $\boldsymbol{U}_i^\top\boldsymbol{U}_i = \boldsymbol{I}_r, i = 1, 2, 3$. Then $\widehat{\mathcal{X}} = \mathcal{D} \times_{i=1}^3 (\boldsymbol{G}_{t,i}^{\frac{1}{2}}\boldsymbol{U}_i)$.

Consider the orthogonalization of $\boldsymbol{G}_{t,i}^{\frac{1}{2}}\boldsymbol{U}_i$:

$$\boldsymbol{G}_{t,i}^{\frac{1}{2}}\boldsymbol{U}_i = \boldsymbol{V}_i\boldsymbol{R}_i, \text{ with } \boldsymbol{V}_i^\top\boldsymbol{V}_i = \boldsymbol{I}_r, \ i = 1, 2, 3.$$

Then

$$\|\boldsymbol{G}_{t,i}^{\frac{1}{2}}\boldsymbol{U}_i\|_F = \|\boldsymbol{V}_i\boldsymbol{R}_i\|_F \leq \|\boldsymbol{R}_i\|_2\|\boldsymbol{V}_i\|_F = \sqrt{r}\|\boldsymbol{R}_i\|_2,$$

thus $\|\boldsymbol{R}_i\|_2 \geq 1/\sqrt{r} \cdot \nu_t^{\frac{1}{6}}\|\boldsymbol{U}_i\|_F = \nu_t^{\frac{1}{6}}$. As the condition number $\kappa(\boldsymbol{R}_i) \leq \kappa(\boldsymbol{G}_{t,i}^{\frac{1}{2}}) \leq \rho_t^{\frac{1}{6}}$, we have

$$\|\boldsymbol{R}_i^{-1}\|_2 \leq \|\boldsymbol{R}_i\|_2^{-1} \cdot \kappa(\boldsymbol{R}_i) \leq \nu_t^{-\frac{1}{6}}\rho_t^{\frac{1}{6}}.$$

Now we consider the $\|\boldsymbol{V}_i\|_{2,\infty} = \max_i \|\boldsymbol{e}_i^\top\boldsymbol{V}_i\|_2$ where $\boldsymbol{e}_i \in \mathbb{R}^{n \times 1}$ is a vector with $i$-th element be $1$ and others be $0$.

$$\|\boldsymbol{e}_i^\top\boldsymbol{V}_i\|_2 = \|\boldsymbol{e}_i^\top\boldsymbol{G}_{t,i}^{\frac{1}{2}}\boldsymbol{U}_i\boldsymbol{R}_i^{-1}\|_2 \leq \|\boldsymbol{e}_i^\top\boldsymbol{G}_{t,i}^{\frac{1}{2}}\boldsymbol{U}_i\|_2 \cdot \|\boldsymbol{R}_i^{-1}\|_2 \leq \nu_t^{-\frac{1}{6}}\rho_t^{\frac{1}{6}}\|\boldsymbol{e}_i^\top\boldsymbol{G}_{t,i}^{\frac{1}{2}}\boldsymbol{U}_i\|_2 \leq \rho_t^{\frac{1}{3}}\|\boldsymbol{e}_i^\top\boldsymbol{U}_i\|_2.$$

So $\|\boldsymbol{V}_i\|_{2,\infty} \leq \rho_t^{\frac{1}{3}}\|\boldsymbol{U}_i\|_{2,\infty}$ and $Incoh(\widehat{\mathcal{X}}) \leq \rho_t^{\frac{1}{3}}\mu$. $\qquad\square$

## Proof of Lemma 4.2

*Proof.* By Lemma B.2, the with the requirement of sampling ratio $p$, the tensor $\mathcal{W}^0$ from Algorithm 2 satisfies

$$\|\mathcal{W}^0 - \mathcal{X}_*\|_F \leq \frac{\sigma_{\min}(\mathcal{X}_*)}{15000C_0r^{\frac{1}{2}}}$$

with high probability. Thus by Lemma B.1, the initial tensor $\mathcal{X}^0 = \mathscr{H}_{\boldsymbol{r}}(\mathrm{Trim}_{\xi}(\mathcal{W}^0))$ with $\xi = \frac{8\|\mathcal{W}^0\|_F}{7n^{3/2}}$ is and $\mathcal{X}^0$ is incoherent with parameter $2\nu\kappa_0$ and satisfies

$$\|\mathcal{X}^0 - \mathcal{X}_*\|_F \leq \|\mathcal{W}^0 - \mathcal{X}_*\|_F + \frac{C_0r^{1/2}\|\mathcal{W}^0 - \mathcal{X}_*\|_F^2}{\sigma_{\min}(\mathcal{X}_*)} \leq \frac{\sigma_{\min}(\mathcal{X}_*)}{10000C_0r^{\frac{1}{2}}}.$$

$\qquad\square$

## Proof of Theorem 4.1

*Proof.* Now we prove the linear convergence of Algorithm 1. Using the idea of induction, we start the proof assuming $\|\mathcal{X}_t - \mathcal{X}_*\|_F \leq \frac{\sigma_{\min}(\mathcal{X}_*)}{10000C_0r^{\frac{1}{2}}}$ and $Incoh(\mathcal{X}_t) \leq 2\nu\kappa_0$ (here the constant $C_0$ keeps the same with constant in Lemma B.1).

Denote $\widehat{\mathcal{X}}_*^t := \mathscr{W}_t^{\frac{1}{2}}\mathcal{X}_*$ and $\alpha_t = \epsilon_t^{\frac{1}{2}}p^{-1}$. Then, by Lemma 3.4, we have

$$\|\mathcal{W}^t - \mathcal{X}_*\|_{\mathscr{W}_t} = \|\mathcal{X}^t - \alpha_t \widetilde{\mathscr{P}}_{\mathbb{T}_{\mathcal{X}^t}} \mathscr{W}_t^{-1} \mathscr{P}_{\Omega}(\mathcal{X}^t - \mathcal{X}_*) - \mathcal{X}_*\|_{\mathscr{W}_t}$$

$$= \|(\widehat{\mathcal{X}}^t - \widehat{\mathcal{X}}_*^t) - \alpha_t \mathscr{P}_{\mathbb{T}_{\widehat{\mathcal{X}}^t}} \mathscr{W}_t^{-\frac{1}{2}} \mathscr{P}_{\Omega} \mathscr{W}_t^{-\frac{1}{2}} (\widehat{\mathcal{X}}^t - \widehat{\mathcal{X}}_*^t)\|_F$$

$$= \|(\widehat{\mathcal{X}}^t - \widehat{\mathcal{X}}_*^t) - \mathscr{P}_{\mathbb{T}_{\widehat{\mathcal{X}}^t}}(\widehat{\mathcal{X}}^t - \widehat{\mathcal{X}}_*^t) - \epsilon_t^{\frac{1}{2}} \mathscr{P}_{\mathbb{T}_{\widehat{\mathcal{X}}^t}}(p^{-1} \mathscr{W}_t^{-\frac{1}{2}} \mathscr{P}_{\Omega} \mathscr{W}_t^{-\frac{1}{2}} - \epsilon_t^{-\frac{1}{2}} \mathscr{I})(\widehat{\mathcal{X}}^t - \widehat{\mathcal{X}}_*^t)\|_F$$

$$\leq \|(\mathscr{I} - \mathscr{P}_{\mathbb{T}_{\widehat{\mathcal{X}}^t}})(\widehat{\mathcal{X}}^t - \widehat{\mathcal{X}}_*^t)\|_F + \epsilon_t^{\frac{1}{2}} \|\mathscr{P}_{\mathbb{T}_{\widehat{\mathcal{X}}^t}}(p^{-1} \mathscr{W}_t^{-\frac{1}{2}} \mathscr{P}_{\Omega} \mathscr{W}_t^{-\frac{1}{2}} - \epsilon_t^{-\frac{1}{2}} \mathscr{I})(\widehat{\mathcal{X}}^t - \widehat{\mathcal{X}}_*^t)\|_F$$

$$= \|(\mathscr{I} - \mathscr{P}_{\mathbb{T}_{\widehat{\mathcal{X}}^t}})(\widehat{\mathcal{X}}^t - \widehat{\mathcal{X}}_*^t)\|_F + \epsilon_t^{\frac{1}{2}} \|\mathscr{P}_{\mathbb{T}_{\widehat{\mathcal{X}}^t}}(p^{-1} \mathscr{W}_t^{-\frac{1}{2}} \mathscr{P}_{\Omega} \mathscr{W}_t^{-\frac{1}{2}} - \mathscr{W}_t^{-1} + \mathscr{W}_t^{-1} - \epsilon_t^{-\frac{1}{2}} \mathscr{I})(\widehat{\mathcal{X}}^t - \widehat{\mathcal{X}}_*^t)\|_F \quad (10)$$

$$\leq \underbrace{\left\|(\mathscr{I} - \mathscr{P}_{\mathbb{T}_{\widehat{\mathcal{X}}^t}})(\widehat{\mathcal{X}}^t - \widehat{\mathcal{X}}_*^t)\right\|_F}_{I_1} + \epsilon_t^{\frac{1}{2}} \underbrace{\left\|\mathscr{P}_{\mathbb{T}_{\widehat{\mathcal{X}}^t}} \mathscr{W}_t^{-\frac{1}{2}}(\mathscr{I} - p^{-1} \mathscr{P}_{\Omega})(\mathcal{X}^t - \mathcal{X}_*)\right\|_F}_{I_2}$$

$$+ \epsilon_t^{\frac{1}{2}} \underbrace{\left\|\mathscr{P}_{\mathbb{T}_{\widehat{\mathcal{X}}^t}}(\mathscr{W}_t^{-1} - \epsilon_t^{-\frac{1}{2}} \mathscr{I})(\widehat{\mathcal{X}}^t - \widehat{\mathcal{X}}_*^t)\right\|_F}_{I_3}.$$

Now we turn to estimate the $I_1, I_2, I_3$, respectively.

**Estimation of $I_1$.**

By Lemma B.5

$$I_1 \leq \frac{7}{\sigma_{\min}(\widehat{\mathcal{X}}_*^t)} \left\|\widehat{\mathcal{X}}^t - \widehat{\mathcal{X}}_*^t\right\|_F^2 \leq \frac{7\rho_t \epsilon_t^{\frac{1}{4}}}{\sigma_{\min}(\mathcal{X}_*)} \left\|\mathcal{X}^t - \mathcal{X}_*\right\|_F^2. \quad (11)$$

where the second inequality follows from (8) and $\sigma_{\min}(\mathscr{W}_t^{\frac{1}{2}} \mathcal{X}_*) \geq \epsilon_t^{\frac{1}{4}} \sigma_{\min}(\mathcal{X}_*)$.

**Estimation of $I_3$.**

By Lemma B.8:

$$I_3 \leq \left\|(\mathscr{W}_t^{-1} - \epsilon_t^{-\frac{1}{2}} \mathscr{I})(\widehat{\mathcal{X}}^t - \widehat{\mathcal{X}}_*^t)\right\|_F$$

$$\leq \epsilon_t^{-\frac{1}{2}} \left(1 - \left(1 + \frac{\|\mathcal{G}_t\|_{\vee}^2}{\epsilon_t}\right)^{-\frac{1}{2}}\right) \left\|\widehat{\mathcal{X}}^t - \widehat{\mathcal{X}}_*^t\right\|_F \quad (12)$$

$$\leq \epsilon_t^{-\frac{1}{4}} \rho_t^{\frac{1}{2}} \left(1 - \left(1 + \frac{\|\mathcal{G}_t\|_{\vee}^2}{\epsilon_t}\right)^{-\frac{1}{2}}\right) \left\|\mathcal{X}^t - \mathcal{X}_*^t\right\|_F,$$

where the third inequality follows from the Lemma B.7.

**Estimation of $I_2$.**

For simplicity, we omit the subscript $t$ and denote $\mathcal{X} = \mathcal{X}^t, \widehat{\mathcal{X}} = \widehat{\mathcal{X}}^t, \widehat{\mathcal{X}}_* = \widehat{\mathcal{X}}_*^t = \mathscr{W}_t^{\frac{1}{2}} \mathcal{X}_*$ in the following. Now we first fix a orthogonal decomposition of $\widehat{\mathcal{X}} = \mathcal{D} \times_{i=1}^3 \widehat{U}_i$, then we choose an orthogonal decomposition of $\widehat{\mathcal{X}}_*$ accordingly. Let $\widehat{\mathcal{X}}_* = \mathcal{D}' \times_{i=1}^3 \widehat{U}_i'$ be a orthogonal decomposition, then we define the following $R_1, R_2, R_3$:

$$R_i = \arg\min_{R \in \mathbb{O}_{r_i}} \|\widehat{U}_i - \widehat{U}_i' R\|_F, \quad i = 1, 2, 3.$$

Here, $\mathbb{O}_{r_i}$ contains all unitary matrices with dimension $r_i \times r_i$. Then we define $\widehat{U}_i^* = \widehat{U}_i' R_i, i = 1, 2, 3$ and $\mathcal{D}_* = \mathcal{D}' \times_{i=1}^3 R_i^\top$. Thus $\widehat{\mathcal{X}}_* = \mathcal{D}_* \times_{i=1}^3 \widehat{U}_i^*$. In this manner, by Lemma B.6 we have:

$$\|\widehat{U}_i - \widehat{U}_i^*\|_F \leq \|U_i U_i^\top - \widehat{U}_i'(\widehat{U}_i')^\top\|_F \leq \frac{\sqrt{2}\|\widehat{\mathcal{X}} - \widehat{\mathcal{X}}_*\|_F}{\sigma_{\min}(\widehat{\mathcal{X}}_*)}, \quad (13)$$

where
$$\mathcal{X} = \mathscr{W}_t^{-\frac{1}{2}}\widehat{\mathcal{X}} = \mathcal{D} \times_{i=1}^3 \widetilde{U}_i, \quad \mathcal{X}_* = \mathscr{W}_t^{-\frac{1}{2}}\widehat{\mathcal{X}}_* = \mathcal{D}_* \times_{i=1}^3 \widetilde{U}_i^*.$$

Then, $I_2$ is rewritten as
$$I_2 = \langle \mathscr{P}_{\mathbb{T}_{\widehat{\mathcal{X}}}} \mathscr{W}_t^{-\frac{1}{2}}(\mathscr{I} - p^{-1}\mathscr{P}_\Omega)(\mathcal{X} - \mathcal{X}_*), \mathcal{X}_0 \rangle$$
$$= \langle (\mathscr{I} - p^{-1}\mathscr{P}_\Omega)(\mathcal{X} - \mathcal{X}_*), \mathscr{W}_t^{-\frac{1}{2}}\mathscr{P}_{\mathbb{T}_{\widehat{\mathcal{X}}}}\mathcal{X}_0 \rangle.$$

for some $\mathcal{X}_0$ with $\|\mathcal{X}_0\|_F \leq 1$. Since $\mathscr{P}_{\mathbb{T}_{\widehat{\mathcal{X}}}}\mathcal{X}_0 = \mathcal{C} \times_{i=1}^3 \widehat{U}_i + \sum_{i=1}^3 \mathcal{D} \times_{j\neq i} \widehat{U}_j \times_i \widehat{W}_i$ for some $\mathcal{C}$ and $\widehat{W}_i, i = 1, 2, 3$, we have
$$\mathscr{W}_t^{-\frac{1}{2}}\mathscr{P}_{\mathbb{T}_{\widehat{\mathcal{X}}}}\mathcal{X}_0 = \underbrace{\mathcal{C} \times_{i=1}^3 \widetilde{U}_i}_{\mathcal{X}_{0,0}} + \sum_{i=1}^3 \underbrace{\mathcal{D} \times_{j\neq i} \widetilde{U}_j \times_i \widetilde{W}_i}_{\mathcal{X}_{0,i}}, \tag{14}$$

where $\mathcal{X}_{0,i}, i = 0, 1, 2, 3$ are all rank-$r$ tensors. We also split $\mathcal{X} - \mathcal{X}_*$ into the sum of rank-$r$ tensors:
$$\mathcal{D} \times_{i=1}^3 \widetilde{U}_i - \mathcal{D}_* \times_{i=1}^3 \widetilde{U}_i^* = \underbrace{(\mathcal{D} - \mathcal{D}_*) \times_1 \widetilde{U}_1 \times_2 \widetilde{U}_2 \times_3 \widetilde{U}_3}_{\mathcal{Y}_0} + \underbrace{\mathcal{D}_* \times_1 (\widetilde{U}_1 - \widetilde{U}_1^*) \times_2 \widetilde{U}_2 \times_3 \widetilde{U}_3}_{\mathcal{Y}_1}$$
$$+ \underbrace{\mathcal{D}_* \times_1 \widetilde{U}_1^* \times_2 (\widetilde{U}_2 - \widetilde{U}_2^*) \times_3 \widetilde{U}_3}_{\mathcal{Y}_2} + \underbrace{\mathcal{D}_* \times_1 \widetilde{U}_1^* \times_2 \widetilde{U}_2^* \times_3 (\widetilde{U}_3 - \widetilde{U}_3^*)}_{\mathcal{Y}_3}. \tag{15}$$

Thus,
$$\langle (\mathscr{I} - p^{-1}\mathscr{P}_\Omega)(\mathcal{X} - \mathcal{X}_*), \mathscr{W}_t^{-\frac{1}{2}}\mathscr{P}_{\mathbb{T}_{\widehat{\mathcal{X}}}}\mathcal{X}_0 \rangle = \sum_{k=0}^3 \sum_{l=0}^3 \langle (\mathscr{I} - p^{-1}\mathscr{P}_\Omega)\mathcal{Y}_k, \mathcal{X}_{0,l} \rangle. \tag{16}$$

Before we estimate the terms in (16) using Lemma B.3, we give the following estimation on factor matrices and core tensors.

- *Upper bound of $\mathcal{C}$ and $\widetilde{W}_i$.*
  - For $\mathcal{C}$: Since $\mathcal{C} = \mathcal{X}_0 \times_{i=1}^3 \widehat{U}_i^\top$ and $\widehat{U}_i$'s are orthogonal matrices, $\|\mathcal{C}\|_F = \|\mathcal{X}_0\|_F \leq 1$.
  - For $\widetilde{W}_i$: Because $\widehat{W}_i = (I_{n_i} - \widehat{U}_i\widehat{U}_i^\top)\mathcal{M}_i(\mathcal{X}_0 \underset{j\neq i}{\times} \widehat{U}_j^\top)\mathcal{M}_i^\dagger(\mathcal{D})$ where $\mathcal{M}_i^\dagger(\mathcal{D})$ is the Moore-Penrose pseudoinverse of $\mathcal{M}_i(\mathcal{D})$ with $\mathcal{M}_i(\mathcal{D})\mathcal{M}_i^\dagger(\mathcal{D}) = I_{r_i}$, we have
    $$\|\widehat{W}_i\|_F \leq \|I_{n_i} - \widehat{U}_i\widehat{U}_i^\top\|_2 \|\mathcal{M}_i^\dagger(\mathcal{D})\|_2 \|\mathcal{X}_0 \times_{j\neq i} \widehat{U}_j^\top\|_F \leq \sigma_{\min}^{-1}(\widehat{\mathcal{X}}) \leq \nu_t^{-\frac{1}{2}}\sigma_{\min}^{-1}(\mathcal{X}),$$
    and thus $\|\widetilde{W}_i\|_F \leq \nu_t^{-\frac{1}{6}}\|\widehat{W}_i\|_F \leq \nu_t^{-\frac{2}{3}}\sigma_{\min}^{-1}(\mathcal{X})$.

- *Upper bound of $\widetilde{U}_i$, $\widetilde{U}_i^*$, $\mathcal{D} - \mathcal{D}_*$, and $\widetilde{U}_i - \widetilde{U}_i^*$.* For ease of exposition, we suppose the incoherence parameter of $\mathcal{X}$ and $\mathcal{X}_*$ are $\mu_0$ and $\mu_1$, respectively. Then by Lemma B.9, the incoherence parameter of $\widehat{\mathcal{X}}$ and $\widehat{\mathcal{X}}_*$ are $\rho_t^{\frac{3}{3}}\mu_0$ and $\rho_t^{\frac{3}{3}}\mu_1$.

  - For $\widetilde{U}_i$: Since $\|\widehat{U}_i\|_F \leq \sqrt{r_i}$ and $\|\widehat{U}_i\|_{2,\infty} \leq \rho_t^{\frac{1}{3}}\mu_0\sqrt{\frac{r_i}{n_i}}$,
    $$\|\widetilde{U}_i\|_F \leq \nu_t^{-\frac{1}{6}}\sqrt{r_i}, \quad \|\widetilde{U}_i\|_{2,\infty} \leq \mu_0\rho_t^{\frac{1}{3}}\nu_t^{-\frac{1}{6}}\sqrt{\frac{r_i}{n_i}}.$$

  - For $\widetilde{U}_i^*$: Similarly,
    $$\|\widetilde{U}_i^*\|_F \leq \nu_t^{-\frac{1}{6}}\sqrt{r_i}, \quad \|\widetilde{U}_i^*\|_{2,\infty} \leq \mu_1\rho_t^{\frac{1}{3}}\nu_t^{-\frac{1}{6}}\sqrt{\frac{r_i}{n_i}}.$$

  - For $\widetilde{U}_i - \widetilde{U}_i^*$: It follows from (13) that
    $$\|\widetilde{U}_i - \widetilde{U}_i^*\|_F \leq \nu_t^{-\frac{1}{6}}\|\widehat{U}_i - \widehat{U}_i^*\|_F \leq \nu_t^{-\frac{1}{6}}\frac{\sqrt{2}\|\widehat{\mathcal{X}} - \widehat{\mathcal{X}}_*\|_F}{\sigma_{\min}(\widehat{\mathcal{X}}_*)} \leq \nu_t^{-\frac{1}{6}}\rho_t^{\frac{1}{2}}\frac{\sqrt{2}\|\mathcal{X} - \mathcal{X}_*\|_F}{\sigma_{\min}(\mathcal{X}_*)}$$
    $$\|\widetilde{U}_i - \widetilde{U}_i^*\|_{2,\infty} \leq (\mu_0 + \mu_1)\rho_t^{\frac{1}{3}}\nu_t^{-\frac{1}{6}}\sqrt{\frac{r_i}{n_i}}.$$

– For $\mathcal{D}$ and $\mathcal{D}_*$: since $\widehat{\mathcal{X}} = \mathcal{D} \times_{i=1}^3 \widehat{\boldsymbol{U}}_i, \widehat{\mathcal{X}}_* = \mathcal{D}_* \times_{i=1}^3 \widehat{\boldsymbol{U}}_i^*$ are two orthogonal decomposition, one has:

$$\|\mathcal{D}\|_F = \|\widehat{\mathcal{X}}\|_F \leq \sqrt{r}\mu_t^{\frac{1}{2}}\sigma_{\max}(\mathcal{X}), \quad \|\mathcal{D}_*\|_F = \|\widehat{\mathcal{X}}_*\|_F \leq \sqrt{r}\mu_t^{\frac{1}{2}}\sigma_{\max}(\mathcal{X}^*)$$

– For $\mathcal{D} - \mathcal{D}_*$: We have

$$\|\mathcal{D}_* - \mathcal{D}\|_F = \|\widehat{\mathcal{X}}_* \times_{i=1}^3 (\widehat{\boldsymbol{U}}_i^*)^\top - \widehat{\mathcal{X}} \times_{i=1}^3 \widehat{\boldsymbol{U}}_i^\top\|_F$$

$$\leq \|\widehat{\mathcal{X}}_* \times_1 (\widehat{\boldsymbol{U}}_1^* - \widehat{\boldsymbol{U}}_1)^\top \times_2 (\widehat{\boldsymbol{U}}_2^*)^\top \times_3 (\widehat{\boldsymbol{U}}_3^*)^\top\|_F + \|\widehat{\mathcal{X}}_* \times_1 \widehat{\boldsymbol{U}}_1^\top \times_2 (\widehat{\boldsymbol{U}}_2^* - \widehat{\boldsymbol{U}}_2)^\top \times_3 (\widehat{\boldsymbol{U}}_3^*)^\top\|_F$$

$$+ \|\widehat{\mathcal{X}}_* \times_1 \widehat{\boldsymbol{U}}_1^\top \times_2 \widehat{\boldsymbol{U}}_2^\top \times_3 (\widehat{\boldsymbol{U}}_3^* - \widehat{\boldsymbol{U}}_3)^\top\|_F + \|(\widehat{\mathcal{X}}_* - \widehat{\mathcal{X}}) \times_{i=1}^3 \widehat{\boldsymbol{U}}_i^\top\|_F$$

$$\leq \|\widehat{\mathcal{X}}_*\|_F \cdot \left(\|\widehat{\boldsymbol{U}}_1^* - \widehat{\boldsymbol{U}}_1\|_F + \|\widehat{\boldsymbol{U}}_2^* - \widehat{\boldsymbol{U}}_2\|_F + \|\widehat{\boldsymbol{U}}_3^* - \widehat{\boldsymbol{U}}_3\|_F\right) + \|\widehat{\mathcal{X}}_* - \widehat{\mathcal{X}}\|_F$$

$$\leq 3\sqrt{r}\sigma_{\max}(\widehat{\mathcal{X}}_*) \cdot \rho_t^{\frac{1}{2}} \frac{\sqrt{2}\|\mathcal{X} - \mathcal{X}_*\|_F}{\sigma_{\min}(\mathcal{X}_*)} + \|\widehat{\mathcal{X}}_* - \widehat{\mathcal{X}}\|_F$$

$$\leq 4\sqrt{r}\kappa_0\mu_t^{\frac{1}{2}}\rho_t^{\frac{1}{2}}\|\mathcal{X} - \mathcal{X}_*\|_F$$

Now we estimate terms $\langle(\mathscr{I} - p^{-1}\mathscr{P}_\Omega)\mathcal{Y}_k, \mathcal{X}_{0,l}\rangle$ for $k, l = 0, 1, 2, 3$ in (16) as follows.

• For $k = l = 0$, we use Lemma B.3. To this end, we notice that

$$\|\widetilde{\boldsymbol{U}}_1\mathscr{M}_1(\mathcal{D} - \mathcal{D}_*)\|_{2,\infty}\|\widetilde{\boldsymbol{U}}_1\mathscr{M}_1(\mathcal{C})\|_F \leq \|\widetilde{\boldsymbol{U}}_1\|_{2,\infty}\|\mathcal{D} - \mathcal{D}_*\|_F\|\widetilde{\boldsymbol{U}}_1\|_F\|\mathcal{C}\|_F$$

$$\leq \mu_0\rho_t^{\frac{1}{3}}\nu_t^{-\frac{1}{6}}\sqrt{\frac{r_1}{n_1}} \cdot \nu_t^{-\frac{1}{6}}\sqrt{r_1} \cdot 4\sqrt{r}\kappa_0\mu_t^{\frac{1}{2}}\rho_t^{\frac{1}{2}}\|\mathcal{X} - \mathcal{X}_*\|_F \cdot 1$$

$$\leq 4\mu_0 r^{\frac{3}{2}}n_1^{-\frac{1}{2}}\nu_t^{-\frac{1}{3}}\mu_t^{\frac{1}{2}}\rho_t^{\frac{5}{6}}\kappa_0\|\mathcal{X} - \mathcal{X}_*\|_F$$

and, for $i = 2, 3$,

$$\|\widetilde{\boldsymbol{U}}_i\|_{2,\infty}\|\widetilde{\boldsymbol{U}}_i\|_F \leq \mu_0\rho_t^{\frac{1}{3}}\nu_t^{-\frac{1}{6}}\sqrt{\frac{r_i}{n_i}} \cdot \nu_t^{-\frac{1}{6}}\sqrt{r_i} \leq \mu_0 r n_i^{-\frac{1}{2}}\nu_t^{-\frac{1}{3}}\rho_t^{\frac{1}{3}}.$$

Thus, Lemma B.3 implies that, with high probability,

$$|\langle(\mathscr{I} - p^{-1}\mathscr{P}_\Omega)\mathcal{Y}_0, \mathcal{X}_{0,0}\rangle| \leq C_1 \left(\frac{\log^3(n)}{p} + \sqrt{\frac{n\log^5(n)}{p}}\right)\mu_0^3 r^{\frac{7}{2}}\bar{n}^{-\frac{3}{2}}\rho_t^2\kappa_0\nu_t^{-\frac{1}{2}}\|\mathcal{X} - \mathcal{X}_*\|_F \tag{17}$$

$$\leq 0.0001\nu_t^{-\frac{1}{2}}\|\mathcal{X} - \mathcal{X}_*\|_F,$$

as long as

$$p \geq C \cdot \max\left\{\log^3(n)\mu_0^3 r^{\frac{7}{2}}\bar{n}^{-\frac{3}{2}}\rho_t^2\kappa_0, \log^5(n)\mu_0^6 r^7 n\bar{n}^{-3}\rho_t^4\kappa_0^2\right\},$$

where $C > 0$ is a sufficiently large but absolute constant.

• When $k = 0$ and $l = 1, 2, 3$, we use Lemma B.3 again. For $l = 1$, we have

$$\|\widetilde{\boldsymbol{U}}_1\mathscr{M}_1(\mathcal{D} - \mathcal{D}_*)\|_{2,\infty}\|\widetilde{\boldsymbol{W}}_1\mathscr{M}_1(\mathcal{D})\|_F \leq \|\widetilde{\boldsymbol{U}}_1\|_{2,\infty}\|\mathcal{D} - \mathcal{D}_*\|_F\|\widetilde{\boldsymbol{W}}_1\|_F\|\mathcal{D}\|_F$$

$$\leq \mu_0\rho_t^{\frac{1}{3}}\nu_t^{-\frac{1}{6}}\sqrt{\frac{r_i}{n_i}} \cdot 4\sqrt{r}\kappa_0\mu_t^{\frac{1}{2}}\rho_t^{\frac{1}{2}}\|\mathcal{X} - \mathcal{X}_*\|_F \cdot \nu_t^{-\frac{2}{3}}\sigma_{\min}^{-1}(\mathcal{X}) \cdot \sqrt{r}\mu_t^{\frac{1}{2}}\sigma_{\max}(\mathcal{X})$$

$$\leq 4\mu_0 r^{\frac{3}{2}}n_1^{-\frac{1}{2}}\nu_t^{-\frac{5}{6}}\mu_t\rho_t^{\frac{5}{6}}\kappa_0\frac{\sigma_{\max}(\mathcal{X})}{\sigma_{\min}(\mathcal{X})}\|\mathcal{X} - \mathcal{X}_*\|_F.$$

For $l = 2, 3$, we have

$$\|\widetilde{\boldsymbol{U}}_1\mathscr{M}_1(\mathcal{D} - \mathcal{D}_*)\|_{2,\infty}\|\widetilde{\boldsymbol{U}}_1\mathscr{M}_1(\mathcal{D})\|_F \leq \|\widetilde{\boldsymbol{U}}_1\|_{2,\infty}\|\mathcal{D} - \mathcal{D}_*\|_F\|\widetilde{\boldsymbol{U}}_1\|_F\|\mathcal{D}\|_F$$

$$\leq \mu_0\rho_t^{\frac{1}{3}}\nu_t^{-\frac{1}{6}}\sqrt{\frac{r_1}{n_1}} \cdot 4\sqrt{r}\kappa_0\mu_t^{\frac{1}{2}}\rho_t^{\frac{1}{2}}\|\mathcal{X} - \mathcal{X}_*\|_F \cdot \nu_t^{-\frac{1}{6}}\sqrt{r_1} \cdot \sqrt{r}\mu_t^{\frac{1}{2}}\sigma_{\max}(\mathcal{X})$$

$$\leq 4\mu_0 r^2 n_1^{-\frac{1}{2}}\nu_t^{-\frac{1}{3}}\mu_t\rho_t^{\frac{5}{6}}\kappa_0\sigma_{\max}(\mathcal{X})\|\mathcal{X} - \mathcal{X}_*\|_F,$$

and

$$\|\widetilde{\boldsymbol{U}}_i\|_{2,\infty}\|\widetilde{\boldsymbol{W}}_i\|_F \le \mu_0\rho_t^{\frac{1}{3}}\nu_t^{-\frac{1}{6}}\sqrt{\frac{r_i}{n_i}}\cdot\nu_t^{-\frac{2}{3}}\sigma_{\min}^{-1}(\mathcal{X}) \le \mu_0 r^{\frac{1}{2}}n_i^{-\frac{1}{2}}\nu_t^{-\frac{5}{6}}\rho_t^{\frac{1}{3}}\sigma_{\min}^{-1}(\mathcal{X}), \quad i = 2, 3.$$

Thus, it follows from Lemma B.3 that, with the conditions satisfied, for $l = 1, 2, 3$,

$$|\langle(\mathscr{I}-p^{-1}\mathscr{P}_\Omega)\mathcal{Y}_0, \mathcal{X}_{0,l}\rangle| \le C_1\left(\frac{\log^3(n)}{p} + \sqrt{\frac{n\log^5(n)}{p}}\right)\mu_0^3 r^{\frac{7}{2}}\bar{n}^{-\frac{3}{2}}\rho_t^{\frac{5}{2}}\kappa_0\frac{\sigma_{\max}(\mathcal{X})}{\sigma_{\min}(\mathcal{X})}\nu_t^{-\frac{1}{2}}\|\mathcal{X}-\mathcal{X}_*\|_F.$$

Summing it over $l$, we obtain that, with high probability,

$$\sum_{l=1}^{3}|\langle(\mathscr{I}-p^{-1}\mathscr{P}_\Omega)\mathcal{Y}_0, \mathcal{X}_{0,l}\rangle| \le 0.0001\nu_t^{-\frac{1}{2}}\|\mathcal{X}-\mathcal{X}_*\|_F, \tag{18}$$

as long as

$$p \ge C\cdot\max\left\{\log^3(n)\mu_0^3 r^{\frac{7}{2}}\bar{n}^{-\frac{3}{2}}\rho_t^{\frac{5}{2}}\frac{\sigma_{\max}(\mathcal{X})}{\sigma_{\min}(\mathcal{X})}\kappa_0, \log^5(n)\mu_0^6 r^7 n\bar{n}^{-3}\rho_t^5\frac{\sigma_{\max}^2(\mathcal{X})}{\sigma_{\min}^2(\mathcal{X})}\kappa_0^2\right\}.$$

- When $k = 1, 2, 3$ and $l = 0$, the estimation is done similarly by using

$$\begin{aligned}|\langle(\mathscr{I}-p^{-1}\mathscr{P}_\Omega)\mathcal{Y}_1, \mathcal{X}_{0,0}\rangle| &\le C_1\left(\frac{\log^3(n)}{p} + \sqrt{\frac{n\log^5(n)}{p}}\right)\|\widetilde{\boldsymbol{U}}_1\|_{2,\infty}\|\mathcal{C}\|_F\|\widetilde{\boldsymbol{U}}_1-\widetilde{\boldsymbol{U}}_1^*\|_F\|\mathcal{D}_*\|_F\\ &\quad\cdot\|\widetilde{\boldsymbol{U}}_2\|_{2,\infty}\|\widetilde{\boldsymbol{U}}_2\|_F\cdot\|\widetilde{\boldsymbol{U}}_3\|_{2,\infty}\|\widetilde{\boldsymbol{U}}_3\|_F\\ &\le C_1\left(\frac{\log^3(n)}{p} + \sqrt{\frac{n\log^5(n)}{p}}\right)\left(\|\widetilde{\boldsymbol{U}}_1\|_{2,\infty}\|\widetilde{\boldsymbol{U}}_2\|_{2,\infty}\|\widetilde{\boldsymbol{U}}_3\|_{2,\infty}\right)\\ &\quad\cdot\|\widetilde{\boldsymbol{U}}_1-\widetilde{\boldsymbol{U}}_1^*\|_F\|\widetilde{\boldsymbol{U}}_2\|_F\|\widetilde{\boldsymbol{U}}_3\|_F\|\mathcal{C}\|_F\|\mathcal{D}_*\|_F\\ &\le C_1\left(\frac{\log^3(n)}{p} + \sqrt{\frac{n\log^5(n)}{p}}\right)\mu_0^3 r^3\bar{n}^{-\frac{3}{2}}\rho_t^2\kappa_0\nu_t^{-\frac{1}{2}}\|\mathcal{X}-\mathcal{X}_*\|_F,\end{aligned}$$

$$\begin{aligned}|\langle(\mathscr{I}-p^{-1}\mathscr{P}_\Omega)\mathcal{Y}_2, \mathcal{X}_{0,0}\rangle| &\le C_1\left(\frac{\log^3(n)}{p} + \sqrt{\frac{n\log^5(n)}{p}}\right)\|\widetilde{\boldsymbol{U}}_1^*\|_{2,\infty}\|\mathcal{D}_*\|_F\|\widetilde{\boldsymbol{U}}_1\|_F\|\mathcal{C}\|_F\\ &\quad\cdot\|\widetilde{\boldsymbol{U}}_2\|_{2,\infty}\|\widetilde{\boldsymbol{U}}_2-\widetilde{\boldsymbol{U}}_2^*\|_F\cdot\|\widetilde{\boldsymbol{U}}_3\|_{2,\infty}\|\widetilde{\boldsymbol{U}}_3\|_F\\ &\le C_1\left(\frac{\log^3(n)}{p} + \sqrt{\frac{n\log^5(n)}{p}}\right)\left(\|\widetilde{\boldsymbol{U}}_1^*\|_{2,\infty}\|\widetilde{\boldsymbol{U}}_2\|_{2,\infty}\|\widetilde{\boldsymbol{U}}_3\|_{2,\infty}\right)\\ &\quad\cdot\|\widetilde{\boldsymbol{U}}_1\|_F\|\widetilde{\boldsymbol{U}}_2-\widetilde{\boldsymbol{U}}_2^*\|_F\|\widetilde{\boldsymbol{U}}_3\|_F\|\mathcal{C}\|_F\|\mathcal{D}_*\|_F\\ &\le C_1\left(\frac{\log^3(n)}{p} + \sqrt{\frac{n\log^5(n)}{p}}\right)\mu_0^2\mu_1 r^3\bar{n}^{-\frac{3}{2}}\rho_t^2\kappa_0\nu_t^{-\frac{1}{2}}\|\mathcal{X}-\mathcal{X}_*\|_F,\end{aligned}$$

and

$$|\langle(\mathscr{I} - p^{-1}\mathscr{P}_\Omega)\mathcal{Y}_3, \mathcal{X}_{0,0}\rangle| \leq C_1 \left( \frac{\log^3(n)}{p} + \sqrt{\frac{n\log^5(n)}{p}} \right) \|\widetilde{U}_1^*\|_{2,\infty} \|\mathcal{D}_*\|_F \|\widetilde{U}_1\|_F \|\mathcal{C}\|_F$$

$$\cdot \|\widetilde{U}_2^*\|_{2,\infty} \|\widetilde{U}_2\|_F \cdot \|\widetilde{U}_3\|_{2,\infty} \|\widetilde{U}_3 - \widetilde{U}_3^*\|_F$$

$$\leq C_1 \left( \frac{\log^3(n)}{p} + \sqrt{\frac{n\log^5(n)}{p}} \right) \left( \|\widetilde{U}_1^*\|_{2,\infty} \|\widetilde{U}_2^*\|_{2,\infty} \|\widetilde{U}_3\|_{2,\infty} \right)$$

$$\cdot \|\widetilde{U}_1\|_F \|\widetilde{U}_2\|_F \|\widetilde{U}_3 - \widetilde{U}_3^*\|_F \|\mathcal{C}\|_F \|\mathcal{D}_*\|_F$$

$$\leq C_1 \left( \frac{\log^3(n)}{p} + \sqrt{\frac{n\log^5(n)}{p}} \right) \mu_0 \mu_1^2 r^3 \bar{n}^{-\frac{3}{2}} \rho_t^2 \kappa_0 \nu_t^{-\frac{1}{2}} \|\mathcal{X} - \mathcal{X}_*\|_F.$$

Summing them up and noticing $\mu_0 \geq \mu_1$ give that, with high probability,

$$\sum_{k=1}^{3} |\langle(\mathscr{I} - p^{-1}\mathscr{P}_\Omega)\mathcal{Y}_k, \mathcal{X}_{0,0}\rangle| \leq 0.0001 \nu_t^{-\frac{1}{2}} \|\mathcal{X} - \mathcal{X}_*\|_F, \tag{19}$$

as long as

$$p \geq C \cdot \max\left\{ \log^3(n)\mu_0^3 r^3 \bar{n}^{-\frac{3}{2}} \rho_t^2 \frac{\sigma_{\max}(\mathcal{X})}{\sigma_{\min}(\mathcal{X})} \kappa_0, \log^5(n)\mu_0^6 r^6 n\bar{n}^{-3} \rho_t^4 \frac{\sigma_{\max}^2(\mathcal{X})}{\sigma_{\min}^2(\mathcal{X})} \kappa_0^2 \right\}.$$

- When $k \neq 0$, $l \neq 0$, and $k = l$, we further divide it into three cases.

    - For $k = l = 1$, recall that

    $$\mathcal{X}_{0,1} = \mathcal{D} \times_1 \widetilde{W}_1 \times_2 \widetilde{U}_2 \times_3 \widetilde{U}_3, \quad \mathcal{Y}_1 = \mathcal{D}_* \times_1 (\widetilde{U}_i - \widetilde{U}_1^*) \times_2 \widetilde{U}_2 \times_3 \widetilde{U}_3,$$

    which, together with Lemma B.4, implies that

    $$|\langle(\mathscr{I} - p^{-1}\mathscr{P}_\Omega)\mathcal{Y}_1, \mathcal{X}_{0,1}\rangle| \leq C_2 \sqrt{\frac{\mu_0^4 n r^2 \log n}{p\bar{n}^3}} \|\mathcal{X}_{0,1}\|_F \cdot \|\mathcal{Y}_1\|_F$$

    $$\leq C_2 \sqrt{\frac{\mu_0^4 n r^2 \log n}{p\bar{n}^3}} \|\widetilde{U}_2\|_F^2 \|\widetilde{U}_3\|_F^2 \|\mathcal{D}_*\|_F \|\mathcal{D}\|_F \cdot \|\widetilde{W}_1\|_F \cdot \|\widetilde{U}_1 - \widetilde{U}_1^*\|_F$$

    $$\leq C_2 \sqrt{\frac{\mu_0^4 n r^2 \log n}{p\bar{n}^3}} r^3 \rho_t^{\frac{3}{2}} \kappa_0 \frac{\sigma_{\max}(\mathcal{X})}{\sigma_{\min}(\mathcal{X})} \nu_t^{-\frac{1}{2}} \|\mathcal{X} - \mathcal{X}_*\|_F.$$

    - For $k = l = 2$, notice that $\mathcal{X}_{0,2} = \mathcal{D} \times_1 \widetilde{U}_1 \times_2 \widetilde{W}_2 \times_3 \widetilde{U}_3$ and

    $$\mathcal{Y}_2 = \underbrace{\mathcal{D}_* \times_1 \widetilde{U}_1 \times_2 (\widetilde{U}_2 - \widetilde{U}_2^*) \times_3 \widetilde{U}_3}_{\mathcal{Y}_{2,1}} + \underbrace{\mathcal{D}_* \times_1 (\widetilde{U}_1^* - \widetilde{U}_1) \times_2 (\widetilde{U}_2 - \widetilde{U}_2^*) \times_3 \widetilde{U}_3}_{\mathcal{Y}_{2,2}}.$$

    Therefore, Lemma B.4 gives

    $$|\langle(\mathscr{I} - p^{-1}\mathscr{P}_\Omega)\mathcal{Y}_{2,1}, \mathcal{X}_{0,2}\rangle| \leq C_2 \sqrt{\frac{\mu_0^4 n r^2 \log n}{p\bar{n}^3}} \|\mathcal{X}_{0,2}\|_F \cdot \|\mathcal{Y}_2\|_F$$

    $$\leq C_2 \sqrt{\frac{\mu_0^4 n r^2 \log n}{p\bar{n}^3}} \|\widetilde{U}_1\|_F^2 \|\widetilde{U}_3\|_F^2 \|\mathcal{D}_*\|_F \|\mathcal{D}\|_F \cdot \|\widetilde{W}_2\|_F \cdot \|\widetilde{U}_2 - \widetilde{U}_2^*\|_F$$

    $$\leq C_2 \sqrt{\frac{\mu_0^4 n r^2 \log n}{p\bar{n}^3}} r^3 \rho_t^{\frac{3}{2}} \kappa_0 \frac{\sigma_{\max}(\mathcal{X})}{\sigma_{\min}(\mathcal{X})} \nu_t^{-\frac{1}{2}} \|\mathcal{X} - \mathcal{X}_*\|_F,$$

and Lemma B.3 gives

$$|\langle (\mathscr{I} - p^{-1}\mathscr{P}_\Omega)\mathcal{Y}_{2,2}, \mathcal{X}_{0,2}\rangle|$$
$$\le C_1 \left( \frac{\log^3(n)}{p} + \sqrt{\frac{n\log^5(n)}{p}} \right) \|\widetilde{\boldsymbol{U}}_1\|_{2,\infty}\|\mathcal{D}\|_F\|\widetilde{\boldsymbol{U}}_1 - \widetilde{\boldsymbol{U}}_1^*\|_F\|\mathcal{D}_*\|_F$$
$$\cdot \|\widetilde{\boldsymbol{U}}_2 - \widetilde{\boldsymbol{U}}_2^*\|_{2,\infty}\|\widetilde{\boldsymbol{W}}_2\|_F \cdot \|\widetilde{\boldsymbol{U}}_3\|_{2,\infty}\|\widetilde{\boldsymbol{U}}_3\|_F$$
$$\le C_1 \left( \frac{\log^3(n)}{p} + \sqrt{\frac{n\log^5(n)}{p}} \right) \mu_0^2(\mu_0 + \mu_1)r^3\bar{n}^{-\frac{3}{2}}\rho_t^{\frac{5}{2}}\kappa_0 \frac{\sigma_{\max}(\mathcal{X})}{\sigma_{\min}(\mathcal{X})}\nu_t^{-\frac{1}{2}}\|\mathcal{X} - \mathcal{X}_*\|_F.$$

– For $k = l = 3$, we utilize $\mathcal{Y}_3 = \mathcal{D}_* \times_1 \widetilde{\boldsymbol{U}}_1^* \times_2 \widetilde{\boldsymbol{U}}_2^* \times_3 (\widetilde{\boldsymbol{U}}_3 - \widetilde{\boldsymbol{U}}_3^*)$ and

$$\mathcal{X}_{0,3} = \underbrace{\mathcal{D} \times_1 \widetilde{\boldsymbol{U}}_1^* \times_2 \widetilde{\boldsymbol{U}}_2^* \times_3 \widetilde{\boldsymbol{W}}_3}_{\mathcal{X}_{0,3,1}} + \underbrace{\mathcal{D} \times_1 \widetilde{\boldsymbol{U}}_1^* \times_2 (\widetilde{\boldsymbol{U}}_2 - \widetilde{\boldsymbol{U}}_2^*) \times_3 \widetilde{\boldsymbol{W}}_3}_{\mathcal{X}_{0,3,2}} + \underbrace{\mathcal{D} \times_1 (\widetilde{\boldsymbol{U}}_1 - \widetilde{\boldsymbol{U}}_1^*) \times_2 \widetilde{\boldsymbol{U}}_2 \times_3 \widetilde{\boldsymbol{W}}_3}_{\mathcal{X}_{0,3,3}}.$$

Then, it is deducted from Lemma B.4 that

$$|\langle (\mathscr{I} - p^{-1}\mathscr{P}_\Omega)\mathcal{Y}_3, \mathcal{X}_{0,3,1}\rangle| \le C_2 \sqrt{\frac{\mu_1^4 nr^2 \log n}{p\bar{n}^3}}\|\mathcal{X}_{0,3,1}\|_F \cdot \|\mathcal{Y}_3\|_F$$
$$\le C_2 \sqrt{\frac{\mu_1^4 nr^2 \log n}{p\bar{n}^3}}\|\widetilde{\boldsymbol{U}}_1^*\|_F^2\|\widetilde{\boldsymbol{U}}_2^*\|_F^2\|\mathcal{D}_*\|_F\|\mathcal{D}\|_F \cdot \|\widetilde{\boldsymbol{W}}_3\|_F \cdot \|\widetilde{\boldsymbol{U}}_3 - \widetilde{\boldsymbol{U}}_3^*\|_F$$
$$\le C_2 \sqrt{\frac{\mu_1^4 nr^2 \log n}{p\bar{n}^3}}r^3\rho_t^{\frac{3}{2}}\kappa_0 \frac{\sigma_{\max}(\mathcal{X})}{\sigma_{\min}(\mathcal{X})}\nu_t^{-\frac{1}{2}}\|\mathcal{X} - \mathcal{X}_*\|_F,$$

and from Lemma B.3 that

$$|\langle (\mathscr{I} - p^{-1}\mathscr{P}_\Omega)\mathcal{Y}_3, \mathcal{X}_{0,3,2}\rangle| \le C_1 \left( \frac{\log^3(n)}{p} + \sqrt{\frac{n\log^5(n)}{p}} \right) \|\widetilde{\boldsymbol{U}}_1^*\|_{2,\infty}\|\mathcal{D}\|_F\|\widetilde{\boldsymbol{U}}_1^*\|_F\|\mathcal{D}_*\|_F$$
$$\cdot \|\widetilde{\boldsymbol{U}}_2^*\|_{2,\infty}\|\widetilde{\boldsymbol{U}}_2 - \widetilde{\boldsymbol{U}}_2^*\|_F \cdot \|\widetilde{\boldsymbol{U}}_3 - \widetilde{\boldsymbol{U}}_3^*\|_{2,\infty}\|\widetilde{\boldsymbol{W}}_3\|_F$$
$$\le C_1 \left( \frac{\log^3(n)}{p} + \sqrt{\frac{n\log^5(n)}{p}} \right) \mu_1^2(\mu_0 + \mu_1)r^3\bar{n}^{-\frac{3}{2}}\rho_t^{\frac{5}{2}}\kappa_0 \frac{\sigma_{\max}(\mathcal{X})}{\sigma_{\min}(\mathcal{X})}\nu_t^{-\frac{1}{2}}\|\mathcal{X} - \mathcal{X}_*\|_F$$

and

$$|\langle (\mathscr{I} - p^{-1}\mathscr{P}_\Omega)\mathcal{Y}_3, \mathcal{X}_{0,3,3}\rangle| \le C_1 \left( \frac{\log^3(n)}{p} + \sqrt{\frac{n\log^5(n)}{p}} \right) \|\widetilde{\boldsymbol{U}}_1^*\|_{2,\infty}\|\mathcal{D}_*\|_F\|\widetilde{\boldsymbol{U}}_1 - \widetilde{\boldsymbol{U}}_1^*\|_F\|\mathcal{D}\|_F$$
$$\cdot \|\widetilde{\boldsymbol{U}}_2^*\|_{2,\infty}\|\widetilde{\boldsymbol{U}}_2\|_F \cdot \|\widetilde{\boldsymbol{U}}_3 - \widetilde{\boldsymbol{U}}_3^*\|_{2,\infty}\|\widetilde{\boldsymbol{W}}_3\|_F$$
$$\le C_1 \left( \frac{\log^3(n)}{p} + \sqrt{\frac{n\log^5(n)}{p}} \right) \mu_1^2(\mu_0 + \mu_1)r^3\bar{n}^{-\frac{3}{2}}\rho_t^{\frac{5}{2}}\kappa_0 \frac{\sigma_{\max}(\mathcal{X})}{\sigma_{\min}(\mathcal{X})}\nu_t^{-\frac{1}{2}}\|\mathcal{X} - \mathcal{X}_*\|_F.$$

The above inequalities are added to derive that, with high probability,

$$\sum_{i=1}^3 |\langle (\mathscr{I} - p^{-1}\mathscr{P}_\Omega)\mathcal{Y}_i, \mathcal{X}_{0,i}\rangle| \le 0.0001\nu_t^{-\frac{1}{2}}\|\mathcal{X} - \mathcal{X}_*\|_F, \tag{20}$$

as long as

$$p \ge C \cdot \max\left\{ \log^3(n)\mu_0^3 r^3\bar{n}^{-\frac{3}{2}}\rho_t^{\frac{5}{2}}\frac{\sigma_{\max}(\mathcal{X})}{\sigma_{\min}(\mathcal{X})}\kappa_0, \left( \log^5(n)\mu_0^6 r^6 \vee \log(n)\mu_0^2 r^8 \right) n\bar{n}^{-3}\rho_t^5 \frac{\sigma_{\max}^2(\mathcal{X})}{\sigma_{\min}^2(\mathcal{X})}\kappa_0^2 \right\}.$$

- When $k \neq 0, l \neq 0, k \neq l$, we discuss it in several cases.

  - For $k = 1$ and $l = 2, 3$, we have

$$
|\langle (\mathscr{I} - p^{-1} \mathscr{P}_\Omega) \mathcal{Y}_1, \mathcal{X}_{0,2} \rangle| \leq C_1 \left( \frac{\log^3(n)}{p} + \sqrt{\frac{n \log^5(n)}{p}} \right) \|\widetilde{\boldsymbol{U}}_1\|_{2,\infty} \|\mathcal{D}\|_F \|\widetilde{\boldsymbol{U}}_1 - \widetilde{\boldsymbol{U}}_1^*\|_F \|\mathcal{D}_*\|_F
$$
$$
\cdot \|\widetilde{\boldsymbol{U}}_2\|_{2,\infty} \|\widetilde{\boldsymbol{W}}_2\|_F \cdot \|\widetilde{\boldsymbol{U}}_3\|_{2,\infty} \|\widetilde{\boldsymbol{U}}_3\|_F
$$
$$
\leq C_1 \left( \frac{\log^3(n)}{p} + \sqrt{\frac{n \log^5(n)}{p}} \right) \mu_0^3 r^3 \bar{n}^{-\frac{3}{2}} \rho_t^{\frac{5}{2}} \kappa_0 \frac{\sigma_{\max}(\mathcal{X})}{\sigma_{\min}(\mathcal{X})} \nu_t^{-\frac{1}{2}} \|\mathcal{X} - \mathcal{X}_*\|_F
$$

    and

$$
|\langle (\mathscr{I} - p^{-1} \mathscr{P}_\Omega) \mathcal{Y}_1, \mathcal{X}_{0,3} \rangle| \leq C_1 \left( \frac{\log^3(n)}{p} + \sqrt{\frac{n \log^5(n)}{p}} \right) \|\widetilde{\boldsymbol{U}}_1\|_{2,\infty} \|\mathcal{D}\|_F \|\widetilde{\boldsymbol{U}}_1 - \widetilde{\boldsymbol{U}}_1^*\|_F \|\mathcal{D}_*\|_F
$$
$$
\cdot \|\widetilde{\boldsymbol{U}}_2\|_{2,\infty} \|\widetilde{\boldsymbol{U}}_2\|_F \cdot \|\widetilde{\boldsymbol{U}}_3\|_{2,\infty} \|\widetilde{\boldsymbol{W}}_3\|_F
$$
$$
\leq C_1 \left( \frac{\log^3(n)}{p} + \sqrt{\frac{n \log^5(n)}{p}} \right) \mu_0^3 r^3 \bar{n}^{-\frac{3}{2}} \rho_t^{\frac{5}{2}} \kappa_0 \frac{\sigma_{\max}(\mathcal{X})}{\sigma_{\min}(\mathcal{X})} \nu_t^{-\frac{1}{2}} \|\mathcal{X} - \mathcal{X}_*\|_F.
$$

  - For $k = 2$ and $l = 1, 3$, we have

$$
|\langle (\mathscr{I} - p^{-1} \mathscr{P}_\Omega) \mathcal{Y}_2, \mathcal{X}_{0,1} \rangle| \leq C_1 \left( \frac{\log^3(n)}{p} + \sqrt{\frac{n \log^5(n)}{p}} \right) \|\widetilde{\boldsymbol{U}}_1^*\|_{2,\infty} \|\mathcal{D}_*\|_F \|\widetilde{\boldsymbol{W}}_1\|_F \|\mathcal{D}\|_F
$$
$$
\cdot \|\widetilde{\boldsymbol{U}}_2\|_{2,\infty} \|\widetilde{\boldsymbol{U}}_2 - \widetilde{\boldsymbol{U}}_2^*\|_F \cdot \|\widetilde{\boldsymbol{U}}_3\|_{2,\infty} \|\widetilde{\boldsymbol{U}}_3\|_F
$$
$$
\leq C_1 \left( \frac{\log^3(n)}{p} + \sqrt{\frac{n \log^5(n)}{p}} \right) \mu_0^2 \mu_1 r^3 \bar{n}^{-\frac{3}{2}} \rho_t^{\frac{5}{2}} \kappa_0 \frac{\sigma_{\max}(\mathcal{X})}{\sigma_{\min}(\mathcal{X})} \nu_t^{-\frac{1}{2}} \|\mathcal{X} - \mathcal{X}_*\|_F
$$

    and

$$
|\langle (\mathscr{I} - p^{-1} \mathscr{P}_\Omega) \mathcal{Y}_2, \mathcal{X}_{0,3} \rangle| \leq C_1 \left( \frac{\log^3(n)}{p} + \sqrt{\frac{n \log^5(n)}{p}} \right) \|\widetilde{\boldsymbol{U}}_1^*\|_{2,\infty} \|\mathcal{D}_*\|_F \|\widetilde{\boldsymbol{U}}_1\|_F \|\mathcal{D}\|_F
$$
$$
\cdot \|\widetilde{\boldsymbol{U}}_2\|_{2,\infty} \|\widetilde{\boldsymbol{U}}_2 - \widetilde{\boldsymbol{U}}_2^*\|_F \cdot \|\widetilde{\boldsymbol{U}}_3\|_{2,\infty} \|\widetilde{\boldsymbol{W}}_3\|_F
$$
$$
\leq C_1 \left( \frac{\log^3(n)}{p} + \sqrt{\frac{n \log^5(n)}{p}} \right) \mu_0^2 \mu_1 r^3 \bar{n}^{-\frac{3}{2}} \rho_t^{\frac{5}{2}} \kappa_0 \frac{\sigma_{\max}(\mathcal{X})}{\sigma_{\min}(\mathcal{X})} \nu_t^{-\frac{1}{2}} \|\mathcal{X} - \mathcal{X}_*\|_F.
$$

  - For $k = 3$ and $l = 1, 2$, we have

$$
|\langle (\mathscr{I} - p^{-1} \mathscr{P}_\Omega) \mathcal{Y}_3, \mathcal{X}_{0,1} \rangle| \leq C_1 \left( \frac{\log^3(n)}{p} + \sqrt{\frac{n \log^5(n)}{p}} \right) \|\widetilde{\boldsymbol{U}}_1^*\|_{2,\infty} \|\mathcal{D}_*\|_F \|\widetilde{\boldsymbol{W}}_1\|_F \|\mathcal{D}\|_F
$$
$$
\cdot \|\widetilde{\boldsymbol{U}}_2^*\|_{2,\infty} \|\widetilde{\boldsymbol{U}}_2\|_F \cdot \|\widetilde{\boldsymbol{U}}_3\|_{2,\infty} \|\widetilde{\boldsymbol{U}}_3 - \widetilde{\boldsymbol{U}}_3^*\|_F
$$
$$
\leq C_1 \left( \frac{\log^3(n)}{p} + \sqrt{\frac{n \log^5(n)}{p}} \right) \mu_0 \mu_1^2 r^3 \bar{n}^{-\frac{3}{2}} \rho_t^{\frac{5}{2}} \kappa_0 \frac{\sigma_{\max}(\mathcal{X})}{\sigma_{\min}(\mathcal{X})} \nu_t^{-\frac{1}{2}} \|\mathcal{X} - \mathcal{X}_*\|_F
$$

and

$$|\langle(\mathscr{I} - p^{-1}\mathscr{P}_\Omega)\mathcal{Y}_3, \mathcal{X}_{0,2}\rangle| \le C_1 \left( \frac{\log^3(n)}{p} + \sqrt{\frac{n\log^5(n)}{p}} \right) \|\widetilde{\boldsymbol{U}}_1^*\|_{2,\infty}\|\mathcal{D}_*\|_F\|\widetilde{\boldsymbol{U}}_1\|_F\|\mathcal{D}\|_F$$

$$\cdot \|\widetilde{\boldsymbol{U}}_2^*\|_{2,\infty}\|\widetilde{\boldsymbol{W}}_2\|_F \cdot \|\widetilde{\boldsymbol{U}}_3\|_{2,\infty}\|\widetilde{\boldsymbol{U}}_3 - \widetilde{\boldsymbol{U}}_3^*\|_F$$

$$\le C_1 \left( \frac{\log^3(n)}{p} + \sqrt{\frac{n\log^5(n)}{p}} \right) \mu_0\mu_1^2 r^3 \bar{n}^{-\frac{3}{2}} \rho_t^{\frac{5}{2}} \kappa_0 \frac{\sigma_{\max}(\mathcal{X})}{\sigma_{\min}(\mathcal{X})} \nu_t^{-\frac{1}{2}} \|\mathcal{X} - \mathcal{X}_*\|_F.$$

Summing all inequalities up, we obtain that, with high probability,

$$\sum_{k=1}^{3}\sum_{l=1,l\ne k}^{3} |\langle(\mathscr{I} - p^{-1}\mathscr{P}_\Omega)\mathcal{Y}_k, \mathcal{X}_{0,l}\rangle| \le 0.0001\nu_t^{-\frac{1}{2}} \|\mathcal{X} - \mathcal{X}_*\|_F, \tag{21}$$

as long as

$$p \ge C \cdot \max\left\{ \log^3(n)\mu_0^3 r^3 \bar{n}^{-\frac{3}{2}} \rho_t^{\frac{5}{2}} \frac{\sigma_{\max}(\mathcal{X})}{\sigma_{\min}(\mathcal{X})}\kappa_0, \log^5(n)\mu_0^6 r^6 n\bar{n}^{-3} \rho_t^5 \frac{\sigma_{\max}^2(\mathcal{X})}{\sigma_{\min}^2(\mathcal{X})}\kappa_0^2 \right\}.$$

By combining (17), (18), (19), (20), and (21), we have the following estimation of $I_2$, with high probability,

$$I_2 \le \sum_{k=0}^{3}\sum_{l=0}^{3} |\langle(\mathscr{I} - p^{-1}\mathscr{P}_\Omega)\mathcal{Y}_k, \mathcal{X}_{0,l}\rangle| \le 0.0005\nu_t^{-\frac{1}{2}} \|\mathcal{X} - \mathcal{X}_*\|_F, \tag{22}$$

as long as

$$p \ge C \max\left\{ \log^3(n)\mu_0^3 r^{\frac{7}{2}} \bar{n}^{-\frac{3}{2}} \rho_t^{\frac{5}{2}} \frac{\sigma_{\max}(\mathcal{X})}{\sigma_{\min}(\mathcal{X})}\kappa_0, \left(\log^5(n)\mu_0^6 r^7 \vee \log(n)\mu_0^2 r^8\right) n\bar{n}^{-3} \rho_t^5 \frac{\sigma_{\max}^2(\mathcal{X})}{\sigma_{\min}^2(\mathcal{X})}\kappa_0^2 \right\}.$$

Now we take $\epsilon_t = \|\mathcal{G}_t\|_\vee^2$, which yields $\rho_t = \sqrt{2}$. Since $\mu_1 = \kappa_0\nu$, $\mu_0 = 2\nu\kappa_0$, and

$$|\sigma_{\min}(\mathcal{X}^t) - \sigma_{\min}(\mathcal{X}_*)| \vee |\sigma_{\max}(\mathcal{X}^t) - \sigma_{\max}(\mathcal{X}_*)| \le \|\mathcal{X}^t - \mathcal{X}_*^t\|_F \le \frac{1}{10000}\sigma_{\min}(\mathcal{X}_*),$$

the estimates (11)(22)(12) imply that: if

$$p \ge C \cdot \max\left\{ \log^3(n)\nu^3 r^{\frac{7}{2}} \bar{n}^{-\frac{3}{2}} \kappa_0^5, \left(\log^5(n)\nu^6 \kappa_0^6 r^7 \vee \log(n)\nu^2 \kappa_0^2 r^8\right) n\bar{n}^{-3}\kappa_0^4 \right\},$$

then, with high probability, it holds that

$$I_1 \le 0.001\nu^{\frac{1}{2}}\|\mathcal{X}^t - \mathcal{X}_*\|_F, \quad I_2 \le 0.0005\nu_t^{-\frac{1}{2}}\|\mathcal{X}^t - \mathcal{X}_*\|_F, \quad I_3 \le 0.3485\nu_t^{-\frac{1}{2}}\|\mathcal{X}^t - \mathcal{X}_*\|_F.$$

Plugging these estimats in (10), we obtain

$$\|\mathcal{W}^t - \mathcal{X}_*\|_{\mathscr{W}_t} \le I_1 + \epsilon_t^{\frac{1}{2}} \cdot I_2 + \epsilon_t^{\frac{1}{2}} \cdot I_3 \le 0.35\nu_t^{\frac{1}{2}}\|\mathcal{X}^t - \mathcal{X}_*\|_F.$$

Finally, by Lemma B.1, Lemma B.7, and the assumption $\|\mathcal{X}^t - \mathcal{X}_*\|_F \le \frac{\sigma_{\min}(\mathcal{X}_*)}{10000C_0 r^{\frac{1}{2}}}$, we obtain

$$\|\mathcal{X}^{t+1} - \mathcal{X}_*\|_F = \|\mathscr{H}_{\boldsymbol{r}}(\mathrm{Trim}_\zeta(\mathcal{W}^t)) - \mathcal{X}_*\|_F$$

$$\le \|\mathcal{W}^t - \mathcal{X}_*\|_F + \frac{C_0 r^{1/2}\|\mathcal{W}^t - \mathcal{X}_*\|_F^2}{\sigma_{\min}(\mathcal{X}_*)}$$

$$\le \frac{1}{\nu_t^{\frac{1}{2}}}\|\mathcal{W}^t - \mathcal{X}_*\|_{\mathscr{W}_t} + \frac{1}{\nu_t}\frac{C_0 r^{1/2}\|\mathcal{W}^t - \mathcal{X}_*\|_{\mathscr{W}_t}^2}{\sigma_{\min}(\mathcal{X}_*)}$$

$$\le 0.35\|\mathcal{X}^t - \mathcal{X}_*\|_F + \frac{C_0 r^{1/2}}{\sigma_{\min}(\mathcal{X}_*)}0.13\|\mathcal{X}^t - \mathcal{X}_*\|_F^2$$

$$\le 0.351\|\mathcal{X}^t - \mathcal{X}_*\|_F.$$

This implies that $\|\mathcal{X}^{t+1} - \mathcal{X}_*\|_F \leq \frac{\sigma_{\min}(\mathcal{X}_*)}{10000 C_0 r^{\frac{1}{2}}}$ and $Incoh(\mathcal{X}^{t+1}) \leq 2\nu\kappa_0$, which conclude the proof of Theorem 4.1. $\quad\square$

# C. Appendix

## Phase Transition

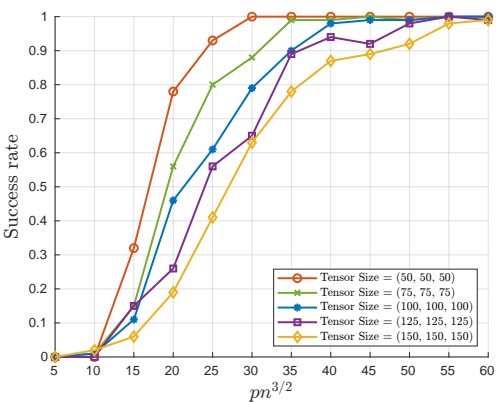 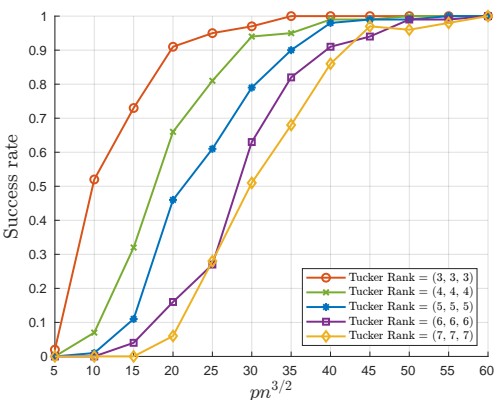

*Figure 6.* The success rate of PRGD with rank $r = 5$ and size $n$ ranging from 50 to 150.

*Figure 7.* The success rate of PRGD for tensor with size $n = 100$ and rank $r$ ranging from 3 to 7.

## Noisy Data Reconstruction

The results of noisy tensor completion with $n = 100, r = 10, \mathrm{OS} = 10$ and $\sigma \in \{5 \times 10^{-4}, 10^{-3}, 5 \times 10^{-3}, 10^{-2}, 5 \times 10^{-2}, 10^{-1}\}$.

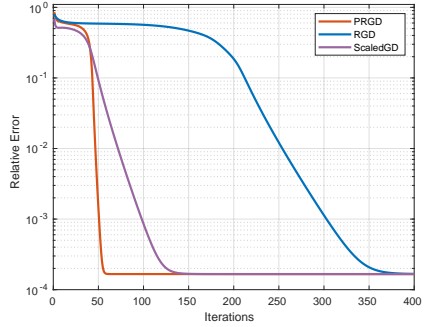 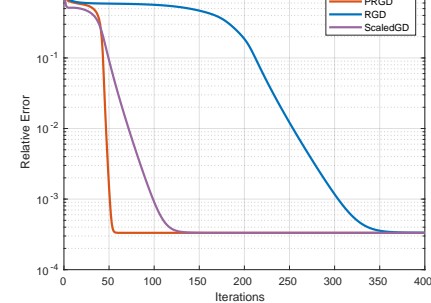 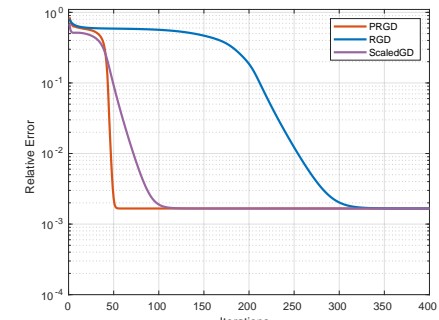

*Figure 8.* $\sigma = 5 \times 10^{-4}$.      *Figure 9.* $\sigma = 10^{-3}$.      *Figure 10.* $\sigma = 5 \times 10^{-3}$.

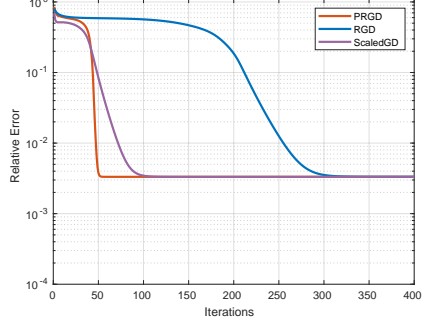 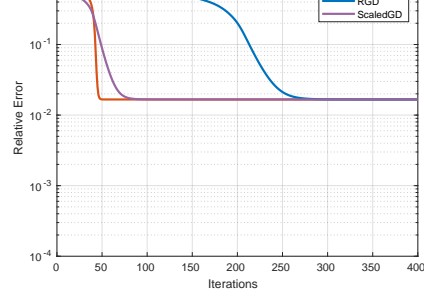 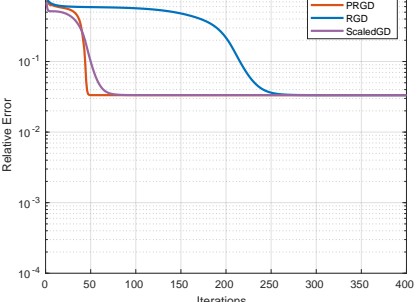

*Figure 11.* $\sigma = 10^{-2}$.      *Figure 12.* $\sigma = 5 \times 10^{-2}$.      *Figure 13.* $\sigma = 10^{-1}$.

**Video Inpainting Result**

The original video is represented as a three-order tensor $\mathcal{X}_*$. For each algorithm, the stopping criteria is chosen when $\|\mathcal{X}^{t+1} - \mathcal{X}^t\|_F / \|\mathcal{X}^t\|_F \leq 10^{-2}$. To demonstrate the representability within the Tucker format, we apply the HOSVD to $\mathcal{X}_*$ with a multilinear rank of $(r, r, r)$ and the resulting tensor $\mathcal{X}_{\mathrm{LR}}$ serves as a near-optimal approximation. Detailed results of recovered tensors from different algorithms, along with $\mathcal{X}_{\mathrm{LR}}$, are presented in Table 2 and Figure 14.

*Table 2.* PSNR values and relative errors of the recovered videos from different algorithms for three test videos under varying multilinear ranks.

| VIDEO(SIZE) | $r$ | PRGD | | RGD | | SCALEDGD | | $\mathcal{X}_{\mathrm{LR}}$ | |
|---|---|---|---|---|---|---|---|---|---|
| | | PSNR | ERR | PSNR | ERR | PSNR | ERR | PSNR | ERR |
| TOMATO | 50 | **27.34** | **8.711E-02** | 27.15 | 8.906E-02 | 27.09 | 8.970E-02 | *27.69* | *8.370E-02* |
| | 60 | **27.79** | **8.272E-02** | 27.44 | 8.608E-02 | 27.15 | 8.900E-02 | *28.41* | *7.699E-02* |
| (242, 320, 501) | 70 | **28.25** | **7.847E-02** | 27.50 | 8.547E-02 | 27.15 | 8.907E-02 | *29.12* | *7.095E-02* |
| HALL MONITOR | 50 | **27.61** | **7.090E-02** | 27.43 | 7.244E-02 | 26.68 | 7.892E-02 | *28.11* | *6.693E-02* |
| | 60 | **28.34** | **6.522E-02** | 28.06 | 6.737E-02 | 26.25 | 8.295E-02 | *29.10* | *5.975E-02* |
| (288, 352, 300) | 70 | **28.83** | **6.166E-02** | 28.75 | 6.220E-02 | 28.52 | 6.390E-02 | *30.08* | *5.337E-02* |
| AKIYO | 50 | **29.29** | **7.616E-02** | 29.09 | 7.787E-02 | 28.81 | 8.045E-02 | *29.87* | *7.123E-02* |
| | 60 | **29.95** | **7.054E-02** | 29.69 | 7.267E-02 | 28.67 | 8.181E-02 | *31.01* | *6.245E-02* |
| (288, 352, 300) | 70 | **30.45** | **6.663E-02** | 30.42 | 6.681E-02 | 28.62 | 8.227E-02 | *32.12* | *5.497E-02* |

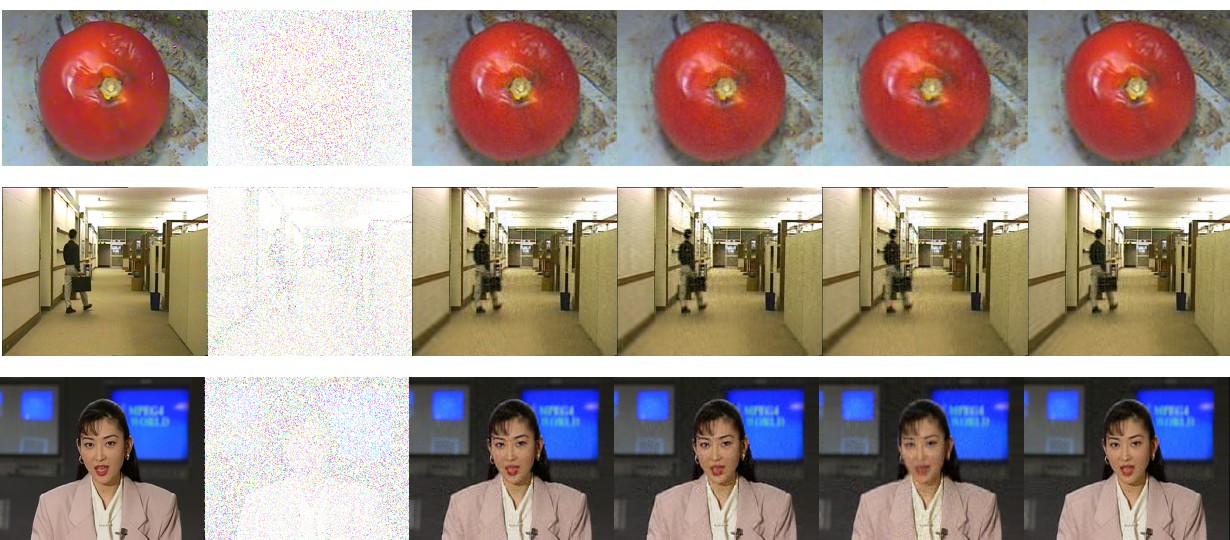

*Figure 14.* The recovered results of three videos with $r = 70$. From top to bottom are *Tomato-1st frame*, *Hall Monitor-40th frame* and *Akiyo-15th frame*. From left to right are *original*, *observed*, *PRGD*, *RGD*, *ScaledGD* and $\mathcal{X}_{\mathrm{LR}}$.

