# OpenReview forum: "Preconditioned Riemannian Gradient Descent Algorithm for Low-Multilinear-Rank Tensor Completion"
_ICML.cc/2025/Conference — ICML 2025 poster_

### Official Review · Reviewer_Mbdu · 2025-03-05

**Overall Recommendation:** 4

**Summary:**

The authors proposa a preconditioned Reimannian gradient descent algorithm for low-rank tensor completion.
The provide analysis of the computational cost and convergence guarantees

**Claims And Evidence:**

The claims are mostly clear and have theorem support.

Question:
- Is $G_{t,i}$ the optimal choice, are there alternatives or tradeoffs to be made?

**Essential References Not Discussed:**

I'm not aware of any misssing key references

**Experimental Designs Or Analyses:**

I think the experimental design is reasonable.

What is the influence of the learning rate on the method's performance? Is the preconditioned Riemannian update sensitive to the curvature of the low-rank manifold?

**Methods And Evaluation Criteria:**

The incorporation of preconditioners to RGD makes sense and evidently improves convergence speed.
Comparisons with state-of-the-art algorithms like RGD and ScaledGD are appropriate.

**Other Comments Or Suggestions:**

see above

**Other Strengths And Weaknesses:**

Strength

-The paper presents a novel approach to tensor completion with clear theoretical and empirical support.

-The algorithm shows significant improvements in convergence speed, which is a critical aspect of optimization algorithms.

Weaknesses

- The paper could benefit from a discussion on the scalability of the PRGD algorithm for very large-scale tensors.

**Questions For Authors:**

- The paper focuses on Tucker decomposition. Can PRGD be extended to Tensor Train (TT) or Tensor Ring decompositions?
- Can the method be generalized to arbitrary high order tensors?


-How sensitive is PRGD to misspecified rank selection?

-Riemannian optimization methods such as Gauss-Newton achieve faster convergence. How does PRGD compare in terms of convergence rates when tested against second-order methods?

- How does the PRGD algorithm handle tensors with very high dimensionality or sparsity? Are there any specific challenges or adjustments needed in such cases?

-Could you provide more insights into the computational complexity of PRGD compared to RGD, especially for large-scale problems?

**Relation To Broader Scientific Literature:**

The paper seems to be well situated within the literature of low-rank tensor completion, and refers to recent works.

**Theoretical Claims:**

The Theorems are well structured and concise.

In lemma 4.2: Can you comment on the practicality of the spectral initialization? What happens if the singular values decay slowly (or at a slower than expected rate)?

---

> ### Author Rebuttal · Authors · 2025-03-31
>
> > **Q1:** The paper focuses on Tucker decomposition. Can PRGD be extended to Tensor Train (TT) or Tensor Ring decompositions?
>
> **A1:**  Thank you for raising this important question. **Please refer to our response to Q2 of Reviewer eeW6.**
>
> > **Q2:** Can the method be generalized to arbitrary high-order tensors?
>
> **A2:** Thank you for your insightful question. **Please refer to our response to Q1 of Reviewer eeW6.**
>
> > **Q3:** How sensitive is PRGD to misspecified rank selection?
>
> **A3:** Thank you for your question. To evaluate the sensitivity of PRGD to rank selection, we conducted additional experiments comparing PRGD with other SOTA algorithms under varying ranks. We fix $n=150,\operatorname{OS}=15$ and choose rank $r$ from $3,4,5,6$.  For each algorithm, we conduct $3$ random trials and count the average iteration number and CPU time (seconds) required to achieve a relative error tolerance $10^{-4}$.
>
> **Iteration number:**
> ||$r=3$|$r=4$|$r=5$|$r=6$|
> |-|-|-|-|-|
> |**PRGD**|71.0|131.3|106.7|53.7|
> |**ScaledGD**|269.0|476.7|296.3|205.0|
> |**RGD**|623.7|2293.0|1697.3|722.0|
>
> **CPU time (seconds):**
> ||$r=3$|$r=4$|$r=5$|$r=6$ |
> |-|-|-|-|-|
> |**PRGD**|1.67|3.16|3.01|1.50|
> |**ScaledGD**|6.45|11.75|7.93|4.92|
> |**RGD**|9.34|35.65|28.45|13.67|
>
> **Key Observation:** PRGD consistently outperforms other algorithms across all tested ranks and is less sensitive to rank selection.
>
> > **Q4:** Riemannian optimization methods such as Gauss-Newton achieve faster convergence. How does PRGD compare in terms of convergence rates when tested against second-order methods?
>
> **A4:** Thank you for your insightful question. Indeed, the Riemannian Gauss-Newton (RGN) algorithm proposed in [1] achieves second-order convergence, while PRGD guarantees only linear convergence.
>
> However, RGN **requires solving a RGN equation at each iteration, which is computationally expensive, particularly for the large-scale problems**. In contrast, the PRGD algorithm maintains a similar computational cost to RGD. Overall,  PRGD might be more computationally efficient than RGN.
>
> [1] Luo, Y, and Zhang. A.R "Low-rank tensor estimation via riemannian gauss-newton: Statistical optimality and second-order convergence." JMLR 24.381 (2023): 1-48.
>
> > **Q5:** How does the PRGD algorithm handle tensors with very high dimensionality or sparsity? Are there any specific challenges or adjustments needed in such cases?
>
> **A5:** Thank you for your insightful question. **The design of PRGD's preconditioners $G_{t, i}$ inherently addresses these concerns**. Specifically, the $G_{t,i}$ in (3) utilizes the outer product of the gradient unfoldings. To ensure computational efficiency in high dimensionality scenarios, we preserve only the diagonal entries, which **reduces the $\mathscr{W}_t^{-1}$ operation to element-wise scaling** of the input tensor. Additionally, those diagonal entries are the norms of mode-$i$ slices of the gradient tensor, thus **computing $G_{t, i}$ takes only $O(|\Omega|)$ operations**. This makes PRGD computationally efficient in the cases of high sparsity.
>
> > **Q6:** Could you provide more insights into the computational complexity of PRGD compared to RGD, especially for large-scale problems?
>
> **A6:** As analysed in lines 237 - 248, the additional computation cost of PRGD is $O(|\Omega| + n r)$, which is negligible compared to the RGD algorithm, whose complexity scales as $O(n^3 r)$. Therefore, PRGD maintains the same computational complexity as RGD for large-scale problems.
>
> > **Q7:** Is $G_{t, i}$ the optimal choice, are there alternatives or tradeoffs to be made?
>
> **A7:** The Hessian matrix would be the optimal choice without considering the computational costs. Our $G_{t,i}$'s approximate the Hessian using the diagonal entries of the outer product of the gradient unfoldings, with $\epsilon_t$ ensuring positive definiteness. This design balances efficiency, leveraging the advantages of tensor products (as in Shampoo [2]) and diagonal approximations (as in AdaGrad [3]), while remaining practical for large-scale problems.
>
> [2] Gupta, V., Koren, T., & Singer, Y. Shampoo: Preconditioned stochastic tensor optimization. In ICML (pp.1842-1850). PMLR, 2018.
>
> [3] Duchi, J, Hazan, E., & Singer, Y. "Adaptive subgradient methods for online learning and stochastic optimization." JMLR 12.7 (2011).
>
> > **Q8:** What is the influence of the learning rate on the method's performance? Is the preconditioned Riemannian update sensitive to the curvature of the low-rank manifold?
>
> **A8:** The learning rate is important for PRGD's performance, as it directly impacts the convergence speed and stability of the algorithm. In our experiments, we tuned the constant step sizes for all tested algorithms to ensure optimal, enabling a fair comparison.
>
> PRGD is not sensitive to the curvature of the low-rank manifold. As demonstrated in Theorem 4.1, the contractive factor is invariant to the condition number of $\mathcal{X}\_*$, which can represent the curvature of the manifold.

---

### Official Review · Reviewer_1EC9 · 2025-03-09

**Overall Recommendation:** 3

**Summary:**

This paper introduces the Preconditioned Riemannian Gradient Descent algorithm for low-multilinear-rank tensor completion, leveraging the manifold structure to achieve faster convergence than standard Riemannian Gradient Descent while maintaining the same per-iteration complexity.

**Claims And Evidence:**

The proposed method appears somewhat unconventional. What is the motivation for introducing (4) and (5)? In line 219, \( W_t \) is not defined. Additionally, if the inverse of \( W \) needs to be computed, it may not be easy or computationally efficient. According to the paper, the proposed method merely adds a complex residual to constrain the gradient’s step size. However, computing this residual is inefficient, and the motivation for using such a residual is unclear.

Why should one transition from classical and simpler methods to the proposed approach, which is more complex but offers only marginal improvements? There should be compelling reasons to persuade researchers to adopt this method.

**Essential References Not Discussed:**

No

**Experimental Designs Or Analyses:**

As shown in Figure 5, the noisy data reconstruction results at different noise levels indicate that the proposed method only outperforms SOTA methods in low-noise scenarios (less than \(10^{-3}\)), which is hardly observable in real-world applications due to the minimal noise presence. However, at higher noise levels (greater than \(10^{-2}\)), which are still relatively moderate compared to real-world cases, the performance advantage is not evident. Providing an explanation for this behavior is essential to substantiate the claimed advantages of the proposed method.

In Fig. 2 and Fig. 3, the text at the top of the figures overlaps with the figures.

**Methods And Evaluation Criteria:**

See claims and evidence.

**Other Comments Or Suggestions:**

None

**Other Strengths And Weaknesses:**

Some of the equations are numbered (e.g., line 23), while others are not (line 45). It is better to number all of them to ensure each equation is trackable. Additionally, all notations should be introduced before they are used, or at the very least, immediately after their first appearance.

**Questions For Authors:**

None

**Relation To Broader Scientific Literature:**

Tensor is widely used in computer vision field.

**Theoretical Claims:**

See claims and evidence.

---

> ### Author Rebuttal · Authors · 2025-03-31
>
> > **Q1:** The proposed method appears somewhat unconventional. What is the motivation for introducing (4) and (5)?
>
> **A1:** Thank you for your question. The derivation of PRGD involves **two essential steps:** (1) endowing the preconditioned metric to the tangent space of the iterate on the manifold. (2) projecting the Euclidean gradient onto the tangent space under the preconditioned metric to obtain the preconditioned Riemannian gradient.
>
> Specifically, **(4) is to define the preconditioned Riemannian metric and (5) is to reparameterize the tangent space**. This reparameterization ensures that the four terms in (5) are mutually orthogonal under the preconditioned metric. Consequently,  we can derive the explicit formula for the tangent space projection operator and the preconditioned Riemannian gradient according to Proposition 3.1.
>
> > **Q2:** In line 219, $\mathscr{W}_t$ is not defined. Additionally, if the inverse of  $\mathscr{W}_t$ needs to be computed, it may not be easy or computationally efficient.
>
> **A2:**  We have defined $\mathscr{W}\_t$ in (4) (line 195) as $\mathscr{W}\_t(\mathcal{Y})=\mathcal{Y}\times\_{i=1}^3G_{t,i}$ for $\mathcal{Y}\in\mathbb{R}^{n_1\times n_2\times n_3}$, where $G_{t,i}\in\mathbb{R}^{n_i\times n_i}$ ($i=1,2,3$) are preconditioned matrices.
>
> Regarding the inverse of $\mathscr{W}\_t$, since $G_{t,i}$ are diagonal matrices, computing $\mathscr{W}\_t^{-1}$ reduces to **element-wise scaling** of the input tensor. In PRGD, we only apply $\mathscr{W}\_t^{-1}$ to the sparse gradient tensor $\mathcal{G}^t=\mathscr{P}\_{\Omega}(\mathcal{X}^t-\mathcal{X}\_*)$ , which only needs $O(|\Omega|)$ operations. **This cost is negligible compared to RGD's computations**. Thus, the $\mathscr{W}_t^{-1}$ is computatioanlly efficient.
>
> > **Q3:** According to the paper, the proposed method merely adds a complex residual to constrain the gradient’s step size. However, computing this residual is inefficient, and the motivation for using such a residual is unclear.
>
> **A3:** Thank you for your comment. We are pleased to clarify the motivation and computation efficiency of our preconditioners in (3) (line 187).
>
> **Computational Efficiency:** Constructing $G_{t,i}$ requires the diagonal entries of the outer product of gradient unfoldings, which correspond to the norms of mode-$i$ slices of gradient tensor (as explained in lines 202-206). Since gradient tensor is sparse, **this computation takes only $O(|\Omega|)$ operations, making it computationally efficient**.
>
> **Motivation:** The outer product of gradient unfolding is used to **approximate the Hessian matrix** of objective function, as demonstrated in conventional optimization techniques such as AdaGrad [1] and Adam. To ensure computational efficiency, we preserve only the diagonal entries, while the parameter $\epsilon_t$ ensures $G_{t,i}$ remains positive definite. Intuitively, this metric flattens the landscape of the objective function, enabling PRGD to take more effective descent directions.
>
> [1] Duchi, J, Hazan, E., & Singer, Y. "Adaptive subgradient methods for online learning and stochastic optimization." JMLR 12.7 (2011).
>
> > **Q4:** Why should one transition from classical and simpler methods to the proposed approach,...,There should be compelling reasons to persuade researchers to adopt this method.
>
> **A4:** Thank you for your question. The PRGD algorithm we proposed is **computationally efficient while offering substantial improvements** over RGD.
>
> **As analyzed in lines 237-248**, the additional cost of preconditioning in PRGD is negligible compared with RGD.  **Thus, PRGD maintains the same per-iteration computational complexity as RGD.**.
>
> More importantly, our numerical experiments (such as Fig 2 and Fig 3) demonstrate that PRGD **achieves approximately $10\times$ acceleration** compared to standard RGD. We believe this improvement is substantial and provides a compelling reason to adopt our method.
>
> > **Q5:** As shown in Fig 5, the noisy data....substantiate the claimed advantages of the proposed method.
>
> **A5:**  Thank you for your question. The noisy data reconstruction experiment is designed to **evaluate the robustness of the algorithm**. As shown in the optimization plots (Page 24, Lines 1285-1319), **our PRGD algorithm consistently converges faster than the other two algorithms** across all noise levels. Consequently, when the stopping criterion ($||\mathcal{X}^{t+1}-\mathcal{X}^t||_F/||\mathcal{X}^t||_F\leq 10^{-4}$) is met, PRGD achieves a lower relative error.
>
> Furthermore, Fig 5 demonstrates PRGD's robustness, the relative error remains consistently below the noise level $\sigma$. This highlights the effectiveness of PRGD and substantiates the claimed advantages.
>
> > **Q6:** In Fig. 2 and Fig. 3, the text... Some of the equations are numbered...their first appearance
>
> **A6**: We sincerely appreciate your valuable feedback. We have corrected all identified typos and made a full proofreading of the final version.

---

### Official Review · Reviewer_eeW6 · 2025-03-14

**Overall Recommendation:** 3

**Summary:**

In this paper, the author introduces a Preconditioned Riemannian Gradient Descent (PRGD) algorithm for low tensor completion based on Tucker decomposition model. A data-driven Riemannian metric is proposed to accelerate convergence. Theoretical analysis is given to guarantee the recovery performance. Experimental results verify the desired performance of the proposed method.

**Claims And Evidence:**

Yes.

**Essential References Not Discussed:**

N/A

**Experimental Designs Or Analyses:**

I've checked the experimental analysis. No big issues.

**Methods And Evaluation Criteria:**

Yes.

**Other Comments Or Suggestions:**

N/A

**Other Strengths And Weaknesses:**

Strength:

Data-driven method with theoretical guarantee and experiment verifications.

Weakness:
1. The order of tensor is limited to 3.
2. Limited comparison in experiments.

**Questions For Authors:**

1. Could the proposed method be applied to higher-order data (4th-order tensor or higher?)
2. Could the proposed method be extended to other types of tensor decomposition model (CP, TT or TR)?
2. Please add comparisons on real data, as tensor completion is a task that many tensor completion methods (not only Tucker format) can be compared.

**Relation To Broader Scientific Literature:**

The author proposes a novel data-driven Riemannian metric construction method with theoretical guarantees, contributing to efficiency in tensor completion.

**Theoretical Claims:**

I roughly check the proofs. No big issues.

---

> ### Author Rebuttal · Authors · 2025-03-31
>
> > **Q1:** Could the proposed method be applied to higher-order data (4th-order tensor or higher?)
>
> **A1:**  Thank you for raising this important question. Indeed, our PRGD algorithm can be extended to the higher-order tensor case and handle higher-order data.
>
> From the algorithmic perspective, for the general $d$ order tensor case, we need firstly to compute preconditioners, e.g. the diagonal matrices $G_{t, i}, i=1, \dots, d$, following the same manner as in (3). And the data-driven metric in (4) will be $\langle\mathcal{X}\times_{i=1}^d G_{t, i}, \mathcal{Y}\rangle$ where $\mathcal{X}, \mathcal{Y}\in \mathbb{R}^{n_1\times \cdots\times n_d}$ are two arbitrary tensor. Next, we can **extend the tangent space parameterization in Lemma 3.2 to the $d$ order case**, which involves $d$ gauge matrices and $d+1$ components as in (5).  Based on this tangent space parameterization, we can derive the tangent space projection formula under $\langle\cdot,\cdot \rangle_{\mathscr{W}_t}$ and the preconditioned Riemannian gradient.
>
> Regarding the convergence theory, we first need to extend our key tools, the concentration inequalities in Lemma B. 2 and Lemma B.3, to the $d$ order case. Once this is done, **the contraction analysis of the distances between iterates and ground truth can be conducted within the same framework** as presented in the current work, thereby establishing the linear convergence of the algorithm.
>
> > **Q2:** Could the proposed method be extended to other types of tensor decomposition model (CP, TT or TR)?
>
> **A2:** Thank you for your insightful observation.  Indeed, the proposed method can be extended to the TT format, as the set of **fixed-TT-rank tensors forms a smooth manifold** [1]. By endowing the tangent space of this manifold with our preconditioned metric, one can derive a TT-PRGD method.
>
> Regarding the CP and TR formats, the situation is less straightforward. To the best of our knowledge, the set of fixed-CP-rank tensors does not form a smooth manifold, and it remains unverified whether the fixed-TR-rank set forms a smooth manifold. However, it is still worth exploring other preconditioning strategies for these formats, which could be an interesting direction for future research.
>
> [1] Holtz, Sebastian, Thorsten Rohwedder, and Reinhold Schneider. "On manifolds of tensors of fixed TT-rank." *Numerische Mathematik* 120.4 (2012): 701-731.
>
>
> >**Q3:** Please add comparisons on real data, as tensor completion is a task that many tensor completion methods (not only Tucker format) can be compared.
>
> **A3:** Thank you for your insightful suggestions. We have expanded our experiments to include comparisons with other format tensor completion methods, specifically the Tensor-Train format RGD (TT-RGD) algorithm [2], on both color image and video datasets. The code of TT-RGD is obtained from the Manopt toolbox. The results demonstrate that **TT-RGD lags in both performance and speed**, highlighting the advantage of the Tucker-rank approach. **Detailed comparison results and analysis are included in our response to Q1 of Reviewer NJ9K.**
>
> [2] Cai, Jian-Feng, Jingyang Li, and Dong Xia. "Provable tensor-train format tensor completion by Riemannian optimization." *Journal of Machine Learning Research* 23.123 (2022): 1-77.

---

### Official Review · Reviewer_NJ9K · 2025-03-23

**Overall Recommendation:** 3

**Summary:**

This paper introduces the Preconditioned Riemannian Gradient Descent (PRGD) algorithm for low-multilinear-rank tensor completion. By designing a data-driven Riemannian metric and an efficient diagonal preconditioner derived from gradient statistics, PRGD achieves 10× faster convergence than standard Riemannian Gradient Descent (RGD) while maintaining comparable computational complexity. Theoretically, PRGD guarantees linear convergence under near-optimal sampling complexity \(O(n^{3/2})\), validated by synthetic experiments (e.g., 10× speedup for \(n=100\) tensors) and real-world video inpainting tasks (e.g., 0.75 dB higher PSNR than RGD on the Tomato dataset at rank 70). The method addresses tensor incoherence/spikiness via gradient-based preconditioning and tangent space parameterization, providing a computationally efficient solution for high-dimensional tensor optimization.

**Claims And Evidence:**

Please refer to below

**Essential References Not Discussed:**

Please refer to below

**Experimental Designs Or Analyses:**

Please refer to below

**Methods And Evaluation Criteria:**

Please refer to below

**Other Comments Or Suggestions:**

There are some typos，eg:
Riemannain optimization" → "Riemannian optimization"，"ap plied" → "applied"，Sampling complexity formula uses \(\overline{n}\) without prior definition (defined later in 1-36), and others.
The author should check typos carefully.

**Other Strengths And Weaknesses:**

Advantage：This manuscript combines preprocessing techniques with Riemannian optimization and proposes a data-driven preprocessing Riemannian metric. In terms of theory, the author provides rigorous proofs demonstrating the linear convergence of the method and that the sampling complexity is close to the theoretical lower bound.
Disadvantages:
1.In the experimental section, the author presents too few experimental results, which do not provide sufficient evidence to convincingly demonstrate the advantages of the proposed algorithm. The author should refer to article and increase the number of experiments. The experiments should include color images and videos, and performance metrics such as PSNR, SSIM, and TIME should be provided. The datasets can be referenced from the following sources: https://sipi.usc.edu/database/database.php and http://trace.eas.asu.edu/yuv/.
2. The author should provide ablation experiments.
3. Preprocessing and Riemannian optimization both appear to be existing technologies. The innovation of PRGD lies more in the combination of these methods rather than a disruptive breakthrough.

**Questions For Authors:**

The author mentioned about data-driven metrics. I'm not sure if data-driven here means being able to automatically learn according to the characteristics of the data. What features does it have?

**Relation To Broader Scientific Literature:**

Please refer to below

**Theoretical Claims:**

Please refer to below

---

> ### Author Rebuttal · Authors · 2025-03-30
>
> > **Q1:** In the experimental section, the author presents too few...should be provided.
>
> **A1:** Thank you for your constructive feedback. We have conducted video inpainting on the videos from your recommended source (see section 5.3 and page 25). In response to your suggestion, we expanded the experimental evaluation to include additional **color image and video inpainting tasks,** incorporating the **suggested PSNR, SSIM, and Runtime (seconds) metrics**. The datasets used are *Tomato* (T), *Akiyo* (A), *Hall-Monitor* (H) videos, as well as the *Airplane (F-16)* color image.
>
> Additionally, as suggested by Reviewer eeW6, we compare our algorithm with TT-RGD (Tensor-Train RGD) on both image and video tasks. Below, we summarize the results:
>
> **Video Results:** Sampling ratio $\rho=0.1$, Tucker rank $(70,70,70)$. For TT-RGD, we set TT rank $(33,33)$ to match parameter dimensionality.
>
> | |PSNR(T)|SSIM(T)|TIME(T)|PSNR(A)|SSIM(A)|TIME(A)|PSNR(H)|SSIM(H)|TIME(H)|
> |-|-|-|-|-|-|-|-|-|-|
> |**PRGD**|**28.25**|**0.754**|12.80|**30.45**|0.869|23.91|**28.83**|**0.824**|39.83|
> |**RGD**|27.50|0.711|9.59|30.42|**0.870**|16.58|28.75|0.820|24.61|
> |**ScaledGD**|27.15|0.678|**9.27**|28.62|0.812|**12.57**|28.52|0.813|**23.98**|
> |**TT-RGD**|26.65|0.673|158.05|28.27|0.821|143.49|25.71|0.736|180.57|
>
> **Image Results:** *Airplane(F-16)* image with size $(512,512,3)$, $\rho=0.3$.  The Tucker rank is set to $(r,r,3)$ and varies $r=20,30,40$. The TT rank $(r_t,3)$ is set accordingly to match parameter dimensionality.
> |$r=20,r_t=10$|PSNR|SSIM|TIME|
> |-|-|-|-|
> |**PRGD**|**23.16**|**0.659**|0.41|
> |**RGD**|22.93|0.648|0.35|
> |**ScaledGD**|22.80|0.647|**0.33**|
> |**TT-RGD**|20.74|0.591|0.83|
>
> |$r=30,r_t=16$|PSNR|SSIM|TIME|
> |-|-|-|-|
> |**PRGD**|**24.80**|**0.695**|0.76|
> |**RGD**|24.71|0.692|0.73|
> |**ScaledGD**|24.70|0.695|**0.69**|
> |**TT-RGD**|22.16|0.609|1.05|
>
> |$r=40,r_t=21$|PSNR|SSIM|TIME|
> |-|-|-|-|
> |**PRGD**|**25.77**|**0.716**|0.83|
> |**RGD**|25.46|0.704|**0.61**|
> |**ScaledGD**|25.47|0.703|**0.61**|
> |**TT-RGD**|22.76|0.612|1.35|
>
> **Key observations:** PRGD consistently outperforms baselines in PSNR/SSIM with comparable runtime needed. Also, TT-RGD lags in both performance and speed, highlighting the advantage of the Tucker-rank approach.
>
> > **Q2:** The author should provide ablation experiments
>
> **A2:** Thank you for your suggestion. Actually, in our current experiments, we have already included key ablation analyses. The primary innovation of PRGD is the preconditioned metric. **By removing this component, PRGD reduces to standard RGD**, which we compare against in all experiments. For real-world data, we further validate PRGD by **testing different rank parameters** and giving detailed comparisons.
> Additionally, we are happy to conduct additional ablation studies if you could specify particular components.
>
> > **Q3:**  Preprocessing and Riemannian optimization both appear to be existing technologies. The innovation of PRGD lies more in the combination of these methods rather than a disruptive breakthrough.
>
> **A3:**  Thank you for your comment. To the best of our knowledge, existing Riemannian optimization methods on the fixed-multilinear-rank manifold use the canonical metric. We design a computationally efficient preconditioner tailored to the fixed-multilinear-rank manifold and provide rigorous proofs of convergence theory, which distinguishes PRGD from prior works.
>
> **Computational efficiency:** Methods like Shampoo [1] rely on dense preconditioners requiring outer products of historical gradients, which are expensive for large-scale problems. **PRGD’s preconditioners are several lightweight diagonal matrices**, preserving similar per-iteration cost as RGD while achieving more than $10\times$ faster speed (as empirically validated).
>
> [1] Gupta, V., Koren, T., & Singer, Y. Shampoo: Preconditioned stochastic tensor optimization. In International Conference on Machine Learning (pp.1842-1850). PMLR, 2018.
>
> >**Q4:** The author mentioned about data-driven metrics. I'm not sure if data-driven here means being able to automatically learn according to the characteristics of the data. What features does it have?
>
> **A4:** Thank you for raising this important question. Our data-driven metric is designed to automatically adapt to the local geometry of the optimization landscape by utilizing gradient information at each iteration. The preconditioners $G_{t, i}$ in (3) are constructed from **the iteration data**, more specifically, the diagonal entries of the outer product of gradient unfoldings, which implicitly approximate the Hessian matrix. This ensures the metric flattens the objective function, enabling PRGD to take more effective descent directions.
> > **Q5**: There are some typos....carefully.
>
> **A5**: We sincerely appreciate your careful reading and valuable feedback. We have corrected all identified typos and made a full proofreading of the final version.

---

### Decision · Program_Chairs · 2025-05-01

**Decision:**

Accept (poster)

**Comment:**

A preconditioned Riemannian gradient descent (RGD) algorithm is proposed for low multilinear rank tensor completion. Recognizing that tensors with a fixed multilinear rank form a smooth embedded submanifold, some standard operations such as projection to a tangent space and retraction (based on HOSVD) are presented for applying Riemannian gradient descent. A preconditioning strategy is proposed to apply in every iteration of RGD. Assuming the initialization is good enough, the proposed algorithm guarantees linear convergence to the global optimum with high probability.